# A deimmunized and pharmacologically optimized Toll-like receptor 5 agonist for therapeutic applications

Vadim Mett[1], Oleg V. Kurnasov[2], Ivan A. Bespalov[3], Ivan Molodtsov[4], Craig M. Brackett[5], Lyudmila G. Burdelya[5], Andrei A. Purmal[3], Anatoli S. Gleiberman[3], Ilia A. Toshkov[3], Catherine A. Burkhart[1], Yakov N. Kogan[3], Ekaterina L. Andrianova[3], Andrei V. Gudkov [3,5✉] & Andrei L. Osterman[2]

The Toll-like receptor 5 (TLR5) agonist entolimod, a derivative of *Salmonella* flagellin, has therapeutic potential for several indications including radioprotection and cancer immunotherapy. However, in Phase 1 human studies, entolimod induced a rapid neutralizing immune response, presumably due to immune memory from prior exposure to flagellated enterobacteria. To enable multi-dose applications, we used structure-guided reengineering to develop a next-generation, substantially deimmunized entolimod variant, GP532. GP532 induces TLR5-dependent NF-κB activation like entolimod but is smaller and has mutations eliminating an inflammasome-activating domain and key B- and T-cell epitopes. GP532 is resistant to human entolimod-neutralizing antibodies and shows reduced *de novo* immunogenicity. GP532 also has improved bioavailability, a stronger effect on key cytokine biomarkers, and a longer-lasting effect on NF-κB. Like entolimod, GP532 demonstrated potent prophylactic and therapeutic efficacy in mouse models of radiation-induced death and tissue damage. These results establish GP532 as an optimized TLR5 agonist suitable for multi-dose therapies and for patients with high titers of preexisting flagellin-neutralizing antibodies.

[1] Buffalo BioLabs, LLC, Buffalo, NY, USA. [2] Sanford Burnham Prebys Medical Discovery Institute, La Jolla, CA, USA. [3] Genome Protection, Inc., Buffalo, NY, USA. [4] Gamaleya Research Center of Epidemiology and Microbiology, Moscow, Russia. [5] Roswell Park Comprehensive Cancer Center, Buffalo, NY, USA. ✉email: andrei.gudkov@roswellpark.org

Innate immunity is a pivotal contributor to human defenses against various exogenous and endogenous stresses and assaults, including microbial infections and cancer, and is also a critical determinant of the efficacy of vaccines. Natural and synthetic agonists of Toll-like receptors (TLRs), which are key activators of innate immune responses, have been considered for numerous therapeutic applications[1]. However, currently only agonists of two out of nine known human TLR family members are FDA-approved: (i) TLR4 agonist monophosphoryl lipid A, used as a vaccine adjuvant, and (ii) TLR7 agonist imiquimod, used topically to treat genital warts, superficial basal cell carcinoma, and actinic keratosis. Agonists of several other TLRs are in various stages of clinical trials, but generally, their development is complicated by substantial safety issues arising from their acute inflammatory toxicities. In this regard, the TLR5 agonist flagellin is a notable exception; the physiological response induced by TLR5 agonists is not only safe (no acute inflammation/cytokine storm) but is also beneficial for a number of medical indications.

Signaling induced by flagellin or a flagellin-derived synthetic TLR5 agonist in TLR5-expressing cells triggers Myd88-dependent activation of transcription factors NF-κB and AP-1. These factors drive expression of numerous target genes, ultimately leading to production of cytokines and other factors with anti-apoptotic, reactive oxygen species (ROS)-scavenging, and tissue-regenerative activities, which together protect normal cells from stresses such as radiation[2–6], ischemia-reperfusion injury[7], graft-versus-host disease[8], etc. Additionally, TLR5 activation has immunostimulatory effects that support use of TLR5 agonists as vaccine adjuvants[9] and for treatment of cancer and inflammatory syndromes[4,10–12]. The biological effects of TLR5 agonists include primary effects on TLR5-expressing target cells and secondary local and systemic effects due to the activity of factors released by target cells.

One pharmacological TLR5 agonist in advanced development is entolimod (previously called CBLB502). Entolimod was engineered from *Salmonella* flagellin by deletion of hypervariable domains D2-D3, which was expected to reduce its immunogenicity[2]. As for flagellin, systemic administration of entolimod causes TLR5-dependent activation of NF-κB leading to production of beneficial cytokines (e.g., G-CSF), but does not induce release of potentially dangerous pro-inflammatory cytokines such as IFN-γ or TNF[3]. The favorable safety profile of entolimod demonstrated in preclinical and clinical studies is presumably due, at least in part, to the fact that functional expression of TLR5 is limited, with liver hepatocytes and intestinal, bladder and other epithelial cells showing the strongest responses to TLR5 agonists[4]. Based on its tissue-protective/regenerative effects, entolimod is being developed as a medical countermeasure against catastrophic radiation exposure[2,5,13]. Entolimod also demonstrated efficacy in preclinical models for a range of medical applications including reduction of the adverse side effects of anticancer radio- and chemotherapies[6,14] and prevention and treatment of liver metastases[4,10,11]. However, it has become clear that these and other clinical opportunities that require repeated drug administration over extended periods of time will likely be hampered by entolimod's neutralizing antigenicity and de novo immunogenicity. While entolimod is ~50-fold less immunogenic than flagellin[2], in clinical trials it was found to trigger a neutralizing antibody response even after a single injection[15]. This rapid immune response likely reflects presence of immune memory cells in nearly all adult humans due to their life-long exposure to flagellated commensal and pathogenic enterobacteria. Moreover, for ~10% of humans, a prohibitively high titer of preexisting neutralizing antibodies (Abs) renders them non-responsive to even a single entolimod injection. These findings indicated that reengineering of entolimod to create a next-generation deimmunized TLR5 agonist was necessary to pursue the multiple opportunities for therapeutic targeting of TLR5.

To meet this challenge, which has only been successfully accomplished for a handful of bacterial proteins (e.g.,[16]), we applied a structure-guided[17] iterative approach to identify and eliminate B- and T-cell epitopes contributing to entolimod's neutralizing antigenicity and immunogenicity. We also aimed to reduce the size of the protein to its minimal functional core (expected to aid deimmunization and bioavailability) and eliminate a domain causing NLRC4 inflammasome activation (expected to further improve safety of the drug by preventing inflammasome-dependent production of pro-inflammatory cytokines such as IL-1β). Here, we describe accomplishment of these goals, resulting in a pharmacologically optimized and substantially deimmunized TLR5 agonist, GP532. GP532 retains near full NF-κB-activating capacity even in the presence of human entolimod-neutralizing Abs, has reduced de novo immunogenicity, and shows therapeutic efficacy as a radiation antidote similar to entolimod in mouse models of lethal total body irradiation (TBI) and sublethal local head and neck (H&N) irradiation. This work represents an important advance towards successful application of TLR5 agonist-based therapy in multiple clinical scenarios.

## Results

**Neutralizing immune response to entolimod in humans.** The TLR5 agonist entolimod has been evaluated in three completed Phase 1 and Phase 2 clinical trials. In these trials, 150 healthy volunteers (no trial registration number assigned) and 25 cancer patients (ClinicalTrials.gov registration number NCT01527136) received from one to four intramuscular (i.m.) or subcutaneous (s.c.) injections of entolimod at doses ranging from 2 to 50 μg per injection. Titers of entolimod-reactive Abs in serum samples collected before and at various times after dosing were measured by ELISA.

Overall, ELISA-determined titers of entolimod-reactive Abs in serum samples correlated with their entolimod-neutralizing activity in a cell-based NF-κB-lacZ reporter assay previously established to quantify TLR5 agonist potency[17]. In the trials with healthy volunteers, 8–10% of participants were found to have preexisting neutralizing entolimod-reactive Abs as indicated by ≥30% inhibition of entolimod activity in NF-κB-lacZ reporter cells by serum collected before entolimod administration (Supplementary Table 1). For the subsequent trial in cancer patients, subjects were prescreened to ensure that neutralizing activity was below this threshold prior to dosing.

In all three trials, serum levels of entolimod-reactive Abs were monitored for various periods of time, up to 11 weeks after the first entolimod injection. This showed that by day 8 post-dose, nearly all study participants developed a strong, sustained increase in entolimod-reactive Ab titers (Fig. 1), with only small decreases observed over the following 6 (Fig. 1b) or even 11 weeks (Fig. 1a). Remarkably, there was no dependence of entolimod-reactive Ab titers on entolimod dose (within the tested dose ranges of 5–10 μg/injection (Fig. 1a) up to 30–40 μg/injection (Fig. 1b)) or number of injections.

The entolimod-reactive Ab response that developed in trial participants following entolimod administration included Abs with entolimod-neutralizing capacity as indicated by the increased proportion of subjects with serum meeting the 30% inhibition threshold in NF-κB-lacz reporter cells. By the last time point (ranging from day 11 to day 30 post-dose in different studies), 94–100% of subjects had neutralizing entolimod-reactive Abs (Supplementary Table 1).

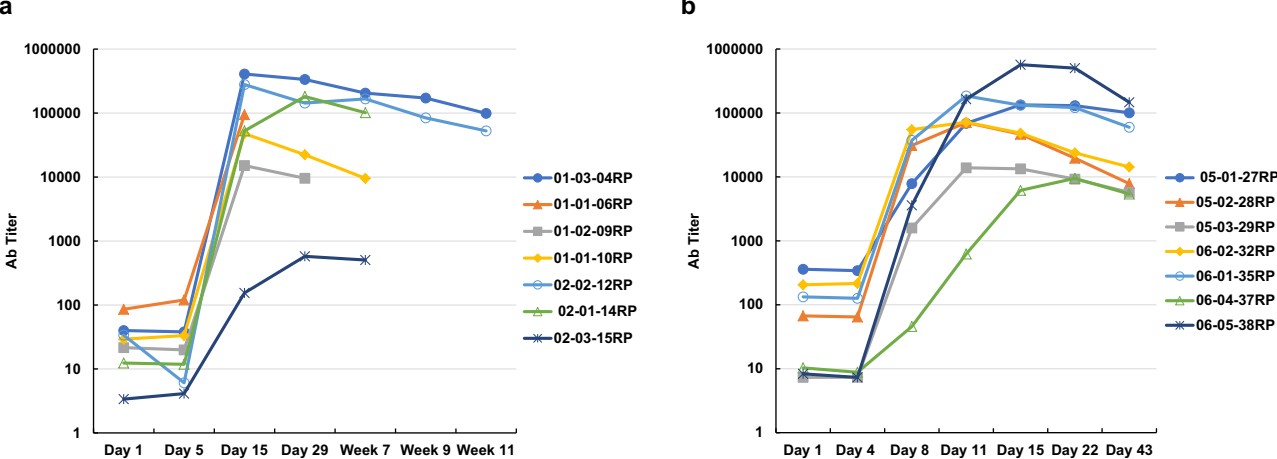

**Fig. 1 Titers of entolimod-reactive antibodies determined by ELISA in serum samples collected from cancer patients administered with different doses of entolimod.** Antibody titer shown for day 1 is for serum collected prior to the first entolimod injection. **a** Entolimod administered daily for 5 days (on days 1–5) s.c. at a dose of 5 µg (subjects 01-03-04RP, 01-01-06RP, 01-02-09RP, and 01-01-10RP) or 10 µg (subjects 02-02-12RP, 02-01-14RP, and 02-03-15RP) per injection per patient. **b** Entolimod administered i.m. (on day 1) and s.c. (on days 4, 8, and 11) at a dose of 30 µg (subjects 05-01-27RP, 05-02-28RP, and 05-03-29RP) or 40 µg (subjects 06-02-32RP, 06-01-35RP, 06-04-37RP, and 06-05-38RP) per injection per patient.

These observations suggest that nearly all adult humans carry anti-flagellin immune memory cells due to prior exposure to flagellated enterobacteria (commensal or pathogenic). These cells appear to direct a potent neutralizing antibody response against entolimod. Thus, a deimmunized version of the drug is needed to treat individuals with high baseline levels of entolimod-reactive Abs and for use in multi-dose therapeutic regimens.

**Engineering of the deimmunized and pharmacologically optimized TLR5 agonist GP532.** To circumvent the limitations imposed by anti-entolimod immune responses observed in human studies, we sought to produce a next-generation deimmunized derivative of entolimod. This involved two distinct tasks. First, we aimed to identify and eliminate the B-cell epitopes in entolimod that contribute to its neutralizing antigenicity. This was expected to minimize the negative impact of preexisting Abs on entolimod's capacity to activate TLR5 signaling and reduce induction of a neutralizing antibody response by anti-flagellin memory cells. Second, we planned to identify and eliminate major T-cell epitopes in entolimod in order to reduce its de novo immunogenicity. While the second task is standard for deimmunization of any therapeutic protein of nonhuman origin, the first is only relevant for exogenous proteins frequently encountered by the human immune system (e.g., bacterial flagellin).

Reengineering of entolimod to generate our final deimmunized variant, GP532, was accomplished in three stages (Fig. 2 and Supplementary Figs. 1 and 2). Intermediate variants were created through truncations or site-directed mutagenesis, expressed in E. coli strain K-12 T7 Express I$^q$ (NEB), and partially purified (to ≥90% homogeneity by SDS-PAGE) on a 5–10 mg scale using our standard entolimod purification protocol (2 M urea extraction from bacterial lysates followed by Ni-NTA mini-column chromatography). All variants were tested for TLR5 agonistic activity using NF-κB signaling in the cell-based lacZ reporter assay as a readout. Only variants with an EC50 ≤5 pM in this assay (compared to ~1 pM characteristic of entolimod) were considered for further reengineering.

*Stage 1: Size reduction.* Here, we aimed to identify the minimal functional core of entolimod required for its TLR5-stimulating activity. Removal of nonessential portions of the protein was

expected to reduce its immunogenicity and potentially improve its bioavailability. In our previous structure–function study[17], deletion of the entire D0 domain (ND0 + CD0) of entolimod substantially reduced activity (up to >300-fold increased EC50 in variant SY3 vs. entolimod, Supplementary Table 2). In this study, deletion of the C-terminal segment of the D0 domain alone also rendered a poorly active TLR5 agonist (>30-fold increased EC50 in variant 470CT vs. entolimod, Supplementary Table 2). In contrast, truncation of the ND0 segment (yielding variant S33) did not impact signaling in NF-κB-lacZ reporter cells (Supplementary Table 2).

Based on these results and analysis of other variants, we pursued further modification of S33. By combining all acceptable deletions, we obtained a Stage 1 final variant named 33ML, which compared to entolimod has (i) size reduced by ~25% (from 329 aa to 250 aa); (ii) N-terminal truncation of the 33 aa-long N-terminal tag and the equally long ND0 segment; (iii) replacement of the 16 aa-long flexible linker (FL) connecting the ND1 and CD1 domains with a 3 aa-long linker; (iv) deletion of 12 N-terminal residues from the CD1 segment; and (v) repositioning of the His$_6$ affinity tag at the C-terminus, linked via a 6 aa-long thrombin cleavage site (Supplementary Fig. 2). 33ML demonstrated TLR5 activation capacity similar to that of entolimod in NF-κB-lacZ reporter cells (EC50 = 1.4 pM) and was used as the starting point for the next stage of deimmunization.

*Stage 2: Mapping and elimination of neutralizing B-cell epitopes.* In this stage, computational tools were applied for (i) prediction of linear B-cell epitopes from entolimod's amino acid sequence (using programs such as Bepipred at http://tools.iedb.org/bcell/) and (ii) prediction of conformational epitopes from the 3D structural model of entolimod (using BEpro at http://pepito.proteomics.ics.uci.edu/). Key residues (typically hydrophilic, charged amino acids) in the most highly ranked predicted epitopes were replaced by alanine within 33ML using site-directed mutagenesis, with typically one to three replacements made at a time (per intermediate variant). Information about flagellin residues essential for high-affinity TLR5 binding and activation obtained in our previous structure–function studies[17] (Supplementary Fig. 3a) was leveraged to preserve functional activity of engineered variants, which was tested in NF-κB-lacZ reporter cells (Supplementary Tables 3 and 4).

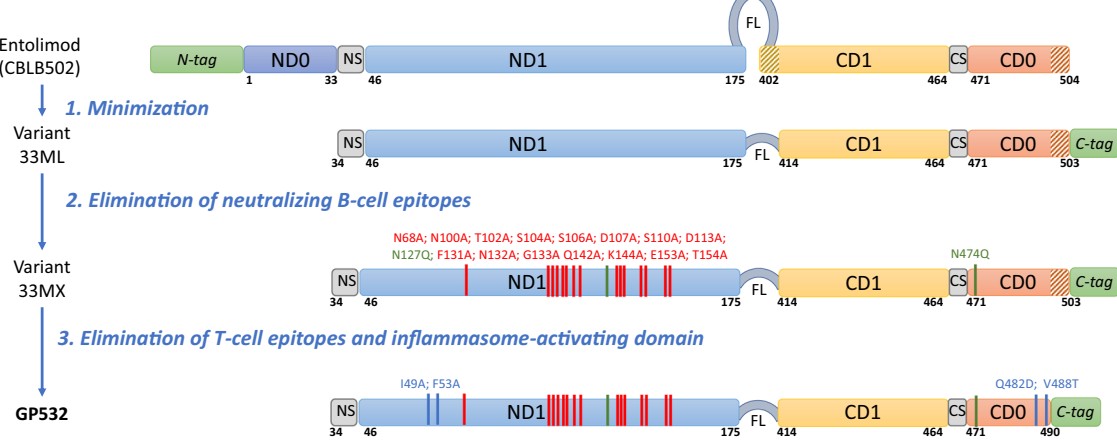

**Fig. 2 Three-stage engineering of the partially deimmunized and pharmacologically optimized TLR5 agonist GP532 from parental entolimod (CBLB502).** N-terminal and C-terminal segments of helical domains D0 and D1 are shown in shades of blue and red/yellow, respectively. The His$_6$ affinity tag added to allow protein purification is shown in green. Unstructured "spoke" regions connecting ND0 with ND1 and CD1 with CD0 are labeled "NS" and "CS", respectively. The flexible linker in entolimod and its minimized version in other variants is labeled "FL". Mutations (alanine substitutions) eliminating B-cell epitopes are listed in red above the ND1 segment, with approximate positions shown by vertical red bars. Green font and vertical bars indicate N→Q substitutions eliminating potential glycosylation sites[35]. Alanine substitutions eliminating two T-cell epitopes in the ND1 and CD0 segments are shown with blue font and vertical bars. A third T-cell epitope (and the inflammasome-activating domain) was eliminated via truncation of the C-terminal 11 amino acids of flagellin (indicated by diagonal hatching).

Next, variants with confirmed NF-κB signaling activity were evaluated for susceptibility to neutralization by performing the same reporter assay in the presence of selected entolimod-specific neutralizing Abs or antisera including serum samples from two patients (P12, P14) who developed the strongest neutralizing Ab response after entolimod administration in a clinical trial (see Supplementary Table 3 for representative variants and Supplementary Table 4 for additional variants). This showed that the main neutralizing B-cell epitopes in 33ML lie within the ND1 helical segment of the D1 domain (Fig. 2). We identified a number of mutations and combinations thereof that had little to no impact on NF-κB signaling induced by the corresponding protein variant in the absence of neutralizing Abs while, at the same time, contributing to resistance of the variant to immunoneutralization (Supplementary Table 3). Ultimately, we assembled 17 such mutations into a single variant, named 33MX (Fig. 3a). Remarkably, this variant displayed NF-κB signaling activity (EC50 = 1.6 pM) that was nearly the same as that of the Stage 1 final variant 33ML (1.4 pM) and very close to that of parental entolimod (1.1 pM), while being ~6–7-fold more resistant to neutralization by the two human antisera samples used in our screening (Supplementary Table 3).

The overall impact of our Stage 2 engineering campaign was assessed by comparing 33MX side-by-side with entolimod and the Stage 1 final variant 33ML in our standard neutralization assay using a set of 45 human serum samples selected based on their intrinsically high titers of neutralizing entolimod-reactive Abs. These serum samples were obtained from anonymous naïve donors (not treated with entolimod) and showed at least 15% inhibition of entolimod activity in NF-κB-lacZ reporter cells (note that 30% inhibition was used as the criterion for presence of neutralizing Abs in entolimod clinical trials, see above). While the neutralization profile of 33ML was essentially the same as that of entolimod (except for several samples falling below 15% neutralization), 33MX showed a significant increase in resistance across the entire panel of antisera (2.5–3-fold decreased inhibition (%) of signaling activity; P < 0.0001 by two-tailed t test; Fig. 3b). Given these findings, we used 33MX as the starting point for

further deimmunization via elimination of T-cell epitopes in Stage 3.

*Stage 3A: Identification and elimination of T-cell epitopes.* In order to suppress entolimod's de novo immunogenicity via elimination of major T-cell epitopes, we utilized the Episcreen$^{TM}$ technology[18] of AbZena (previously Antitope; https://abzena.com), which combines experimental (peptide library screening) and computational (in silico modeling of mapped epitopes to guide their elimination) methods (details below and in Supplementary Methods). Briefly, initial mapping of major T-cell epitopes was performed using a library of 80 synthetic 15 aa peptides covering the entire sequence of 33MX with an overlap of 12 aa between each pair of consecutive peptides. This library was screened for CD4+ T-cell epitopes using peripheral blood mononuclear cell (PBMC) samples from a cohort of 50 donors selected to best represent the number and frequency of HLA-DR and DQ MHC allotypes expressed in the world population. This approach allowed us to at least partially address the challenge that population heterogeneity presents for T-cell epitope analyses. Use of experimental rather than computational screening at this step provides the advantage of quantitative assessment of epitope strength and frequency in the human population while reducing the rate of false positives. Peptides were considered to contain a T-cell epitope if their incubation with PBMCs stimulated both CD4+ T-cell proliferation (measured by [$^3$H]-thymidine incorporation) and IL-2 secretion (measured by ELISpot assay) with signal intensity (SI) ≥1.9 and P < 0.05 for three or more PBMC donors. Based on these criteria, three epitope areas represented by a total of five peptides (Fig. 4) were identified and further analyzed in silico for MHC class II binding core 9-mer HLA-DR restricted epitopes using iTope™ software[19]. This analysis refined putative epitope boundaries, identified anchor residues, and provided guidelines for their elimination via site-directed mutagenesis (see Supplementary Methods, Abzena Reports 1 and 2).

Next, we generated a large set of 33MX-based variants containing 1–3 mutations in identified epitopes as suggested by the iTope program (Supplementary Methods, Abzena Report 2)

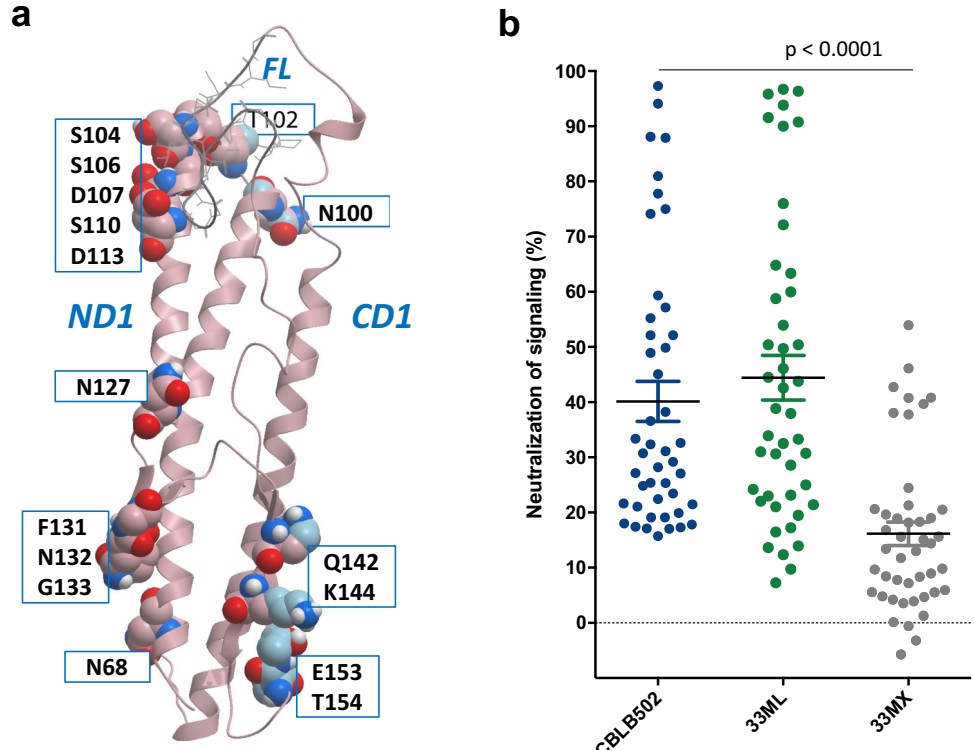

**Fig. 3 Mapping and elimination of neutralizing B-cell epitopes in entolimod. a** 3D model of the D1 domain of entolimod (from the available 3D structure of *Salmonella* flagellin complexed with TLR5 (PDB:3V47[17]), with amino acid residues identified as contributing to neutralizing B-cell epitopes and subjected to site-directed mutagenesis (replacement with alanine) in GP532 shown with spheres over the ribbon diagram (*n* = 15). The potential glycosylation site at N127 that was replaced with alanine in GP532 is also shown. All of these modifications are present in intermediate variant 33MX as well as GP532 (but not in entolimod or variant 33ML). **b** Side-by-side evaluation of entolimod (CBLB502) and two intermediate variants (33ML and 33MX) for inhibition of their signaling activity in HEK293-hTLR5::NF-κB-*lacZ* reporter cells by 45 different human neutralizing antisera. Horizontal lines indicate the mean ± SD (*n* = 45). *P* < 0.0001 for comparison of 33MX to either entolimod or 33ML (ANOVA test). Values below zero likely reflect the extent of assay variability.

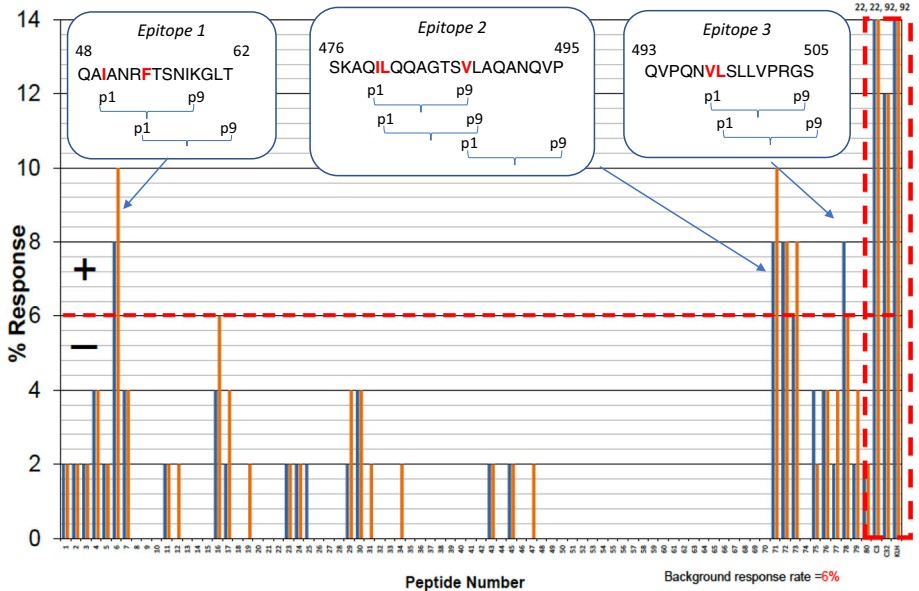

**Fig. 4 Mapping of T-cell epitopes in entolimod.** Non-adjusted (blue bars) and adjusted (orange bars) proliferation assay data for 80 test peptides (numbered 1–80) spanning the sequence of entolimod derivative 33MX and the highly immunogenic control peptides C3, C32, and KLH (dashed box at right end of graph) over a panel of PBMC samples from 50 healthy donors. Peptides inducing positive (SI ≥2.00, *P* < 0.05) T-cell proliferation responses at a frequency above the background response threshold (6%, red dotted line) contain major T-cell epitopes. Peptides spanning identified Epitopes 1–3 and their positions in the 33MX sequence are shown in boxes. Predicted anchor residues (p1) are shown in bold red font. See Supplementary Methods, Abzena Report 1 for additional details.

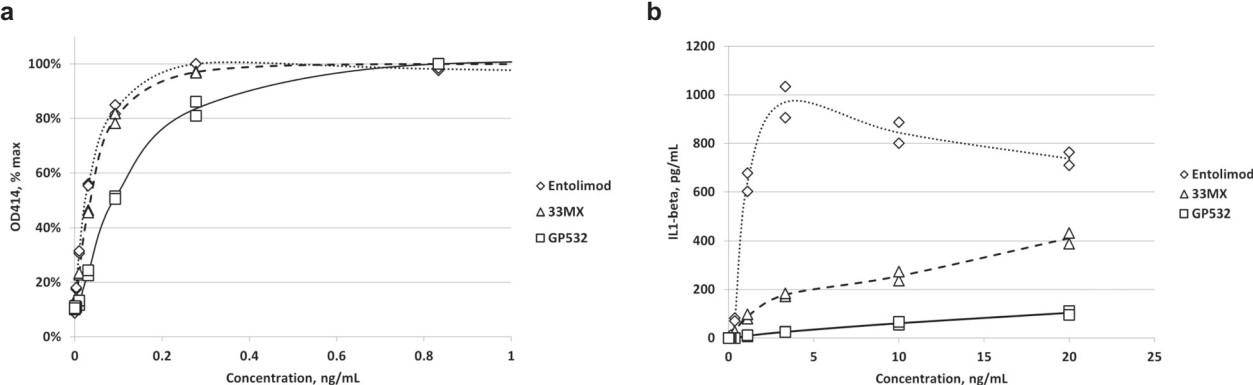

**Fig. 5 Effects of C-terminal capping (in 33MX) and truncation (in GP532) on NF-κB and inflammasome activation by TLR5 agonists. a** NF-κB-dependent induction of lacZ production (as indicated by $OD_{414}$) in HEK293-hTLR5::NF-κB-*lacZ* reporter cells following incubation with different concentrations of entolimod, intermediate variant 33MX and GP532. **b** Inflammasome activation by different concentrations of entolimod, 33MX and GP532 as indicated by IL-1β production in a THP1-NLRC4 cell-based assay (measured by ELISA). Markers indicate individual values for duplicate samples and trend lines pass through mean values.

and tested them for preserved activity in NF-κB-lacZ reporter cells (Supplementary Table 5). Ultimately, we selected two mutations each in Epitopes 1 and 2 and deletion of 11 C-terminal amino acids (Ala492–Leu502, present in flagellin, entolimod and 33MX; Fig. 2 and Supplementary Fig. 1) containing Epitope 3. Combining all of these modifications in GP532 led to only a moderate ~3–4-fold increase in EC50 for in vitro NF-κB signaling compared to 33MX (Supplementary Table 6).

To confirm that the modifications in GP532 did in fact reduce its immunogenicity, we compared GP532 and entolimod using the EpiScreen™ DC:T-cell assay[20] with the same panel of PBMC from 50 healthy donors (performed by AbZena, see Supplementary Methods, Abzena Report 3). Briefly, dendritic cells (DC) derived from monocytes isolated from the PBMC samples were incubated with the test proteins (GP532, entolimod, or the highly immunogenic antigen keyhole limpet hemocyanin (KLH) as a positive control) and induced to mature in order to present T-cell epitopes to autologous purified CD4+ T-cells. T-cell responses were assessed using the same two readouts, proliferation ([³H]-thymidine incorporation) and IL-2 secretion (ELISpot assay). GP532 showed a statistically significant decrease in the frequency of response in both assays as compared to entolimod (3.5-fold and 2-fold, respectively; Supplementary Table 7). An even greater difference was observed when the results of both assays were combined. Thus, while for entolimod 29% of positive responses by the proliferation readout were also positive by ELISpot, not a single case of such agreement between the two assays was observed for GP532 (as a reference, for KLH, correlation between the two readouts was 74%; Supplementary Table 7). Thus, modifications made to generate GP532 substantially reduced its T-cell-directed immunogenicity while not adversely impacting its TLR5 agonistic activity.

*Stage 3B: Elimination of the inflammasome-activating domain.*
The C-terminal deletion that removed Epitope 3 simultaneously accomplished another objective of entolimod reengineering: elimination of its ability to induce inflammasome activation. This "secondary" activity, mechanistically distinct from the primary NF-κB signaling activity, was previously observed for flagellin[21] and confirmed by us for entolimod (Fig. 5 and Supplementary Fig. 4). Since inflammasome activation can induce release of pro-inflammatory cytokines like IL-1β, we hypothesized that elimination of this capacity would improve entolimod's safety. Therefore, in parallel with deimmunization, we sought to

eliminate entolimod residues critical for inflammasome activation, but dispensable for NF-κB signaling.

Prior studies showed that interaction of internalized (or intracellularly expressed) flagellin with the intracellular NAIP5 receptor followed by NLRC4-mediated inflammasome activation is fully determined by flagellin's 35 aa-long CD0 domain[22]. However, we found that deletion of the entire CD0 domain (Thr470-Arg504) drastically reduced its ability to activate NF-κB (variant 470CT, Supplementary Table 2). A more subtle modification, Ala substitution of three C-terminal Leu residues previously shown to be critical determinants of inflammasome activation[22] (L500A; L502A; L503A), did not diminish NF-κB activation by entolimod (variant 502NQ-LA, Supplementary Table 8) but also only moderately suppressed inflammasome activation (Supplementary Fig. 4a). Unexpectedly, we found that capping the entolimod C-terminus with a 12 aa peptide (containing a thrombin cleavage site followed by a His6 tag) combined with deletion of the ND0 domain (as in variants 33ML and 33MX, Supplementary Fig. 2) led to much stronger suppression of inflammasome activation (Supplementary Fig. 4a). After several iterations, we defined an optimal C-terminal deletion (ΔAla492–Leu502) that, within the context of 33MX, completely suppressed inflammasome activation (variant 491MX, Supplementary Fig. 4) but had only a moderate ~2-fold suppressive effect on NF-κB signaling (Supplementary Table 8). This deletion (which also eliminated T-cell Epitope 3, see above) was incorporated into the design of GP532, our final next-generation entolimod derivative. GP532 was shown to retain reasonably high NF-κB signaling activity (EC50 within ~4-fold of entolimod's in NF-κB-lacZ reporter cells; Fig. 5a), with practically no inflammasome activation in vitro (Fig. 5b) or in vivo (Fig. 6c).

**GP532 expression, purification, and characterization.** To produce sufficient GP532 for further studies, our initial small-scale (up to 20 mg) expression and purification protocol was scaled up (by >10-fold), modified, and adjusted to the standards and requirements of the presently launched GP532 industrial manufacturing campaign. The developed batch-feed fermentation process using the same expression strain as previously used for GMP production of entolimod (in fully synthetic media with glycerol) yielded ~1 g/L of GP532 protein extractable by 2 M urea. Further FPLC-based two-step purification consisting of immobilized metal-affinity chromatography (IMAC) and anion exchange chromatography (AEX) was performed on a 200 mg

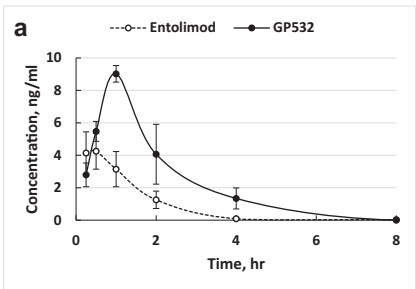 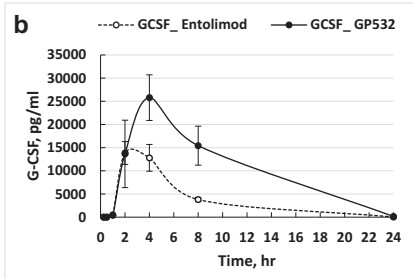 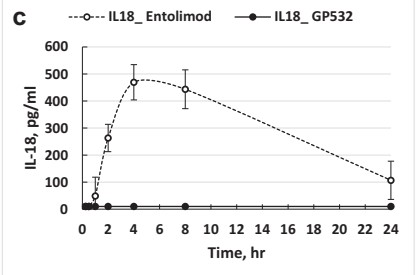

**Fig. 6 Pharmacological characteristics of GP532 and entolimod. a** PK analysis of serum drug concentrations by ELISA after s.c. injection of entolimod and GP532 (1 μg/injection) in C57BL/6 mice (*n* = 5 mice per group per time point). **b**, **c** Serum concentrations of entolimod PD biomarker G-CSF (**b**) and inflammasome biomarker IL-18 (**c**) in the same groups of mice as in (**a**). Mean values ± SD are shown for five mice per group per time point.

scale, yielding ~100% pure protein, comparable to GMP-produced entolimod (Supplementary Fig. 5).

The reduced neutralizing antigenicity of GP532 relative to entolimod was confirmed by comparative inhibitory profiling using 22 normal human serum samples selected from the previously used set of 45 for having the highest degrees of entolimod neutralization (>30%) in NF-κB-LacZ reporter cells (*P* < 0.0001 by ANOVA, Supplementary Fig. 3b). Both median and maximal inhibition (%) of GP532-induced NF-κB signaling by serum samples was >2.5-fold lower than for entolimod. Notably, only 3 out 22 tested serum samples (inhibiting entolimod in the range of 35–95%) inhibited GP532 at or slightly above the 30% cut-off used in entolimod clinical trials.

Side-by-side pharmacokinetic (PK) analysis in C57BL/6 mice (s.c. administration, 1 μg/mouse) demonstrated improved bioavailability of GP532 compared to entolimod (Fig. 6a). Serum concentrations of GP532 peaked at 1 h post-injection vs. 0.5 h for entolimod, the $C_{max}$ value for GP532 was >2-fold higher than for entolimod (9 vs. 4.2 ng/ml), and the AUC value for GP532 was ~3-fold higher than for entolimod (19.2 vs. 6.2 ng × h/L). The exact reason(s) for the improved PK profile of GP532 vs. entolimod remains unknown. Among possible explanations, the smaller size (by ~30%) and higher hydrophobicity (due to replacement of many Asp residues with Ala) of GP532 may contribute to its greater bioavailability and/or sustainability in circulation.

GP532 also displayed an improved pharmacodynamic (PD) profile for previously identified mechanistically-relevant cytokine biomarkers[5], including G-CSF (Fig. 6b). The observed increase in AUC for G-CSF (2.3-fold for GP532 vs. entolimod) likely reflects the comparable increase in PK parameters mentioned above.

Absence of inflammasome activation by GP532 in vivo was confirmed within the same PK/PD study by monitoring IL-18 levels in mouse plasma (Fig. 6c). We monitored this cytokine in lieu of IL-1β (known to be insufficiently abundant in circulation) since NLRC4 inflammasome activation results in caspase-1-dependent proteolytic maturation of both pro-IL-1β and pro-IL-18[23].

Next, we evaluated the dynamics of NF-κB signaling induced by GP532 vs. entolimod in vivo in C57BL/6 mice (s.c., 1 μg/mouse) by immunohistochemical visualization of NF-κB p65 subunit nuclear translocation in liver hepatocytes (known primary responders to TLR5 agonists in mice[4]). While entolimod and GP532 induced comparable p65 nuclear translocation at 30 min post-injection (Fig. 7a, d, respectively), the duration of this effect was substantially longer for GP532, with reduced signal observed at 2 h post-injection for entolimod (Fig. 7b) but not for GP532 (Fig. 7e). No signal was observed for either treatment at 24 h (Fig. 7c, f).

To further confirm the suppressed neutralizing antigenicity of GP532 in vivo, transgenic NF-κB-luciferase reporter mice were

transfused with human entolimod-neutralizing Abs or isotype-matched control Abs and then injected s.c. with entolimod or GP532 (1 μg/mouse). As shown in Fig. 8, live imaging of luciferase expression showed that both entolimod and GP532 induced strong NF-κB signaling in the liver in the absence of neutralizing Abs (see Fig. 8b, d, respectively, compared to vehicle-treated control in Fig. 8a) and that presence of such Abs dramatically suppressed entolimod's activity (Fig. 8c) but had no effect on that of GP532 (Fig. 8e). Overall, our comparative analyses of entolimod and GP532 confirm that GP532 retains comparable NF-κB-stimulating activity while being improved in terms of PK, PD, duration of NF-κB response, lack of inflammasome activation, and resistance to neutralizing Abs both in vitro and in vivo.

**Effects of entolimod and GP532 on transcription.** Since the key physiological effects of TLR5 agonists are mediated by transcriptional responses (i.e., NF-κB and AP-1 activity), we used RNAseq-based analysis of global gene expression in C57BL/6 mice following systemic administration of entolimod vs. GP532 (1 μg/mouse, s.c.) to evaluate the biocomparability of the two drugs. It should be noted that possible effects of TLR5 agonism mediated by inflammasome activation (expected for entolimod, but not GP532) occur post-translationally and, thus, are not observed by RNAseq. In this experiment, we used wild type (WT) and TLR5$^{-/-}$ mice (on the same genetic background) to address the TLR5-dependence/specificity of responses and focused on the liver since it was previously defined as the major organ responsive to TLR5 systemic administration in mice[4]. Livers were collected 30 min or 24 h after drug injection in order to assess primary direct responses and long-term (direct and indirect) effects, respectively.

Comparison of differentially expressed (DE) genes in WT mice at 30 min post-treatment with GP532 or entolimod vs. vehicle showed: (i) For both TLR5 agonists, more genes were upregulated than downregulated (Supplementary Fig. 6a, b). Thus, ~140 genes were significantly upregulated (log2(FC) > 1, adjusted *P* value <0.05) in both entolimod- and GP532-treated groups while ~20 and ~40 genes were downregulated (log2(FC) < −1, adjusted *P* value <0.05) in the same groups, respectively (Supplementary Fig. 6c). This bias is consistent with the known mechanism of action of TLR5 agonists inducing NF-κB-dependent transcription of many genes. (ii) A substantial fraction of upregulated genes (93) and eight downregulated genes were shared between the entolimod- and GP532-treated groups (Supplementary Fig. 6c). This supports the expectation of a common mechanism of action for the two drugs. (iii) The extent of up- or down-regulation within a set of shared DE genes (as reflected by log2(FC) values in pairwise comparison with the vehicle-treated group) showed good correlation between entolimod- and GP532-treated groups (Supplementary Fig. 6d), thus further confirming

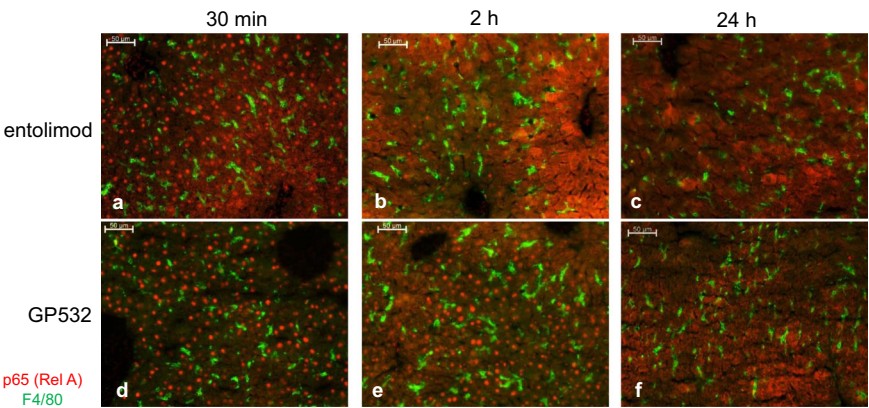

**Fig. 7 Dynamics of NF-κB activation in mouse liver hepatocytes following s.c. injection of entolimod or GP532.** C57BL/6 mice were injected with entolimod (**a**–**c**) or GP532 (**d**–**f**) s.c. (1 µg/injection). Livers were harvested at 30 min (**a**, **d**), 2 h (**b**, **e**), or 24 h (**c**, **f**) post-injection. NF-κB activation was monitored by immunohistochemical visualization of NF-κB p65 subunit nuclear translocation (as in[4]). Red–p65 (RelA); green–F4/80 (macrophage biomarker). Nuclear localization of p65 is indicated by a punctuate staining pattern. Scale bar = 50 µm in all panels.

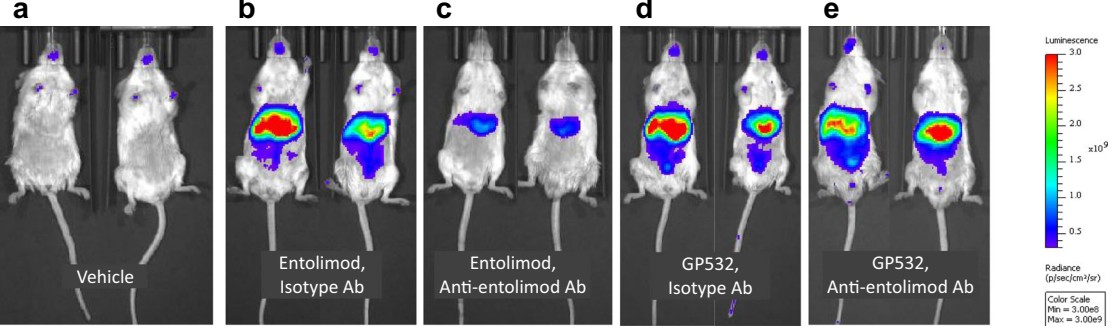

**Fig. 8 Effect of entolimod and GP532 on in vivo NF-κB signaling in reporter mice in the absence or presence of transfused human entolimod-reactive neutralizing antisera.** Transgenic NF-κB-luciferase reporter mice were transfused with neutralizing ("anti-entolimod", panels **c**, **e**) or non-neutralizing ("isotype" control, panels **b**, **d**) human serum followed 2.5 h later by s.c. injection of 1 µg/mouse entolimod (panels **b**, **c**) or GP532 (panels **d**, **e**). Live imaging was performed 3 h post-injection. The intensity of the luminescence signal is indicated by color as shown by the bar to the right of the panels. Control mice were injected s.c. with vehicle only (no transfusion, panel **a**). Images of two representative mice from each group are shown (*n* = 3–4 mice/group).

biocomparability of entolimod and GP532. (iv) Pathway enrichment analysis of DE genes using the KEGG database revealed a shared pattern of pathway enrichment for entolimod and GP532, with the strongest upregulation observed in pathways associated with innate immune responses including TLR and NF-κB signaling (Supplementary Fig. 7).

Data for a set of genes comprising the top five enriched KEGG pathways (from Supplementary Fig. 7; TNF, IL-17, MAPK, and NF-κB signaling pathways and cytokine–cytokine receptor interaction pathway) were used to construct a heatmap (Fig. 9). For simplicity, the heatmap includes only 46 genes showing differential expression (log2(FC) > 0, adjusted *p* value <0.05) between entolimod- and GP532-treated WT mice (lanes A, B) vs. all four control groups (vehicle-treated WT mice and vehicle-, entolimod-, and GP532-treated TLR5$^{-/-}$ mice; lanes C, D, E, and F). This illustrates the consistent patterns of gene upregulation induced by entolimod and GP532. Notably, many of the DE genes in these pathways are known to be directly associated with NF-κB- (*RelA, Ikbke, Nfkbia, Nfkb2, RelB, Myd88*) or AP-1- (*Jund, Jun, Junb*) driven transcription. Additional relevant DE genes revealed by analysis of a larger set of KEGG pathways (Supplementary Fig. 8) included many cytokines, chemokines, and cytokine/chemokine receptors. For all genes identified as TLR5 agonist-responsive, expression was equally low in TLR5$^{-/-}$ mice treated with either agonist and in vehicle-treated WT mice,

thus confirming TLR5-dependence of the observed effects. Consistent with the dynamics of TLR5 agonist-induced signaling (e.g., Figs. 6 and 7), most genes upregulated by entolimod or GP532 at 30 min were no longer upregulated at 24 h (Supplementary Fig. 9). At 24 h, distinct sets of genes (representing different KEGG pathways) were DE, but these showed less pronounced changes and much less bias towards upregulation (Supplementary Figs. 10 and 11). Overall, the effects of entolimod and GP532 on transcription confirmed our expectations of their mechanism of action and general biocomparability.

**Therapeutic efficacy of GP532.** To test the potential clinical usefulness of GP532, we assessed its therapeutic effects in mouse models of three medical indications previously identified as targets for TLR5 agonist-based treatment. First, we evaluated radioprotection in mice exposed to lethal TBI. As shown for entolimod here and in prior studies[2–4], a single dose of GP532 significantly improved survival of NIH Swiss mice when injected (s.c., 1 µg/mouse) 30 min prior to lethal TBI (8.5 Gy) (Fig. 10a). While 100% of vehicle-treated mice died by day 20 post-irradiation, 85% of GP532-treated mice survived to at least day 30, when the study was terminated. Importantly, passive transfer of human entolimod-neutralizing Abs (combined antisera from trial patients administered 1 h before entolimod or GP532) resulted in significant suppression of entolimod's

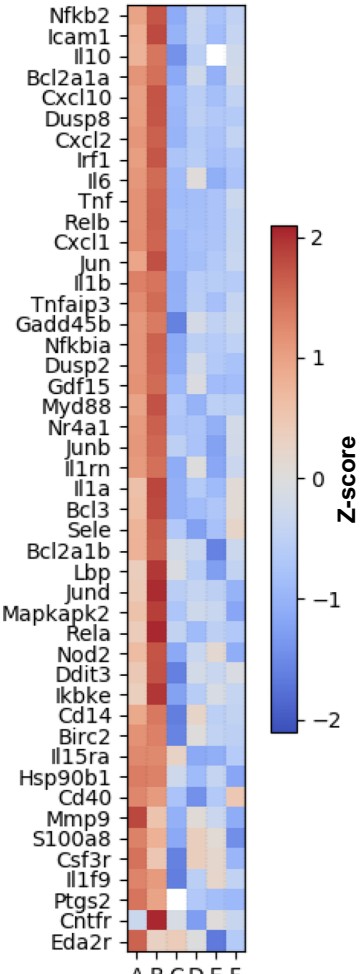

**Fig. 9 Effects of entolimod and GP532 on gene expression in mouse liver.**
RNAseq analysis was performed to identify changes in gene expression in mouse livers collected 30 min after s.c. injection of GP532 (1 μg/mouse, lanes A and D), entolimod (1 μg/mouse, lanes B and E) or vehicle (lanes C and F) in wild type (WT) C57BL/6 mice (lanes A, B, C) or TLR5-KO mice (lanes D, E, F). Data were shown as a heatmap for genes from the top five enriched pathways identified by KEGG pathway analysis: KEGG Pathways mmu04668 (TNF signaling), mmu04657 (IL-17 signaling), mmu04060 (cytokine–cytokine receptor interaction), mmu04010 (MAPK signaling), and mmu04064 (NF-kappa B signaling). These genes were significantly upregulated (deseq2 log2FC > 0, adjusted *P* value < 0.05) in WT mice treated with either entolimod or GP532 as compared to a combined group of vehicle-treated WT mice and all TLR5-KO mice (vehicle-, entolimod-, and GP532-treated). Data were analyzed using deSeq2 R software. deseq2 normalized counts were log2-transformed and Z-score values were calculated independently for each gene (exact zero values were excluded and are shown as white squares). Genes were sorted using an unweighted pair group method with arithmetic mean (UPGMA) hierarchical clustering algorithm. The red-blue colored heatmap shows Z-scores ranging from 2 to −2 as indicated by the vertical bar on the right of the figure.

radioprotective effect (30-day survival reduced from 100 to 20%, *P* = 0.0001), but did not substantially affect that of GP532 (Fig. 10a). This provides proof of the comparable efficacy of GP532 and entolimod for systemic radioprotection and further verification of GP532's resistance to neutralizing Abs.

Next, we compared the ability of GP532 and entolimod to mitigate the lethality of TBI when administered (s.c., 1 μg/mouse in BALB/c mice) 24 h post-irradiation (7.5 Gy TBI). In both drug-

treated groups, mouse survival was significantly higher than in the vehicle-treated group (60% vs. 20% at day 60; *P* = 0.05 and 0.01 for GP532 and entolimod, respectively; Fig. 10b). Thus, the in vivo systemic radiomitigative efficacy of GP532 was comparable to that of entolimod.

Finally, GP532-mediated tissue protection was tested in a mouse model of sublethal local H&N irradiation, which develops damage to mucosal tissues and salivary glands mimicking the side effects of H&N cancer radiotherapy. This model previously demonstrated efficacy of entolimod as a potential treatment for radiation-induced mucositis, a common debilitating side-effect of an otherwise highly effective cancer therapy[24]. Here, we administered GP532 (0.3 μg/mouse, s.c.) to FVB/NJ mice 30 min before single-dose X-ray irradiation (15 Gy) to the H&N area and collected mucosal tissues and salivary glands 30 days later for histopathological analysis. A significant decrease in radiation-induced tissue damage was observed in GP532-treated mice compared to vehicle-treated controls across all analyzed radio-sensitive H&N tissues (Fig. 10c and Supplementary Fig. 12). Thus, GP532 demonstrates potent therapeutic activity comparable to entolimod's in several animal models of tissue damage utilizing different mouse strains and readouts.

## Discussion

Many potential medical applications of flagellin-derived TLR5 agonists cannot be pursued due to their intrinsic immunogenicity. The first-generation TLR5 agonistic drug entolimod (CBLB502[2]) was derived from *Salmonella* flagellin by deletion of its hyper-variable and highly immunogenic D2 and D3 domains. Nevertheless, entolimod retains sufficient neutralizing immunogenicity to trigger a highly inhibitory antibody response within a week after systemic single-dose administration. This was observed in Phase 1 and 2 clinical trials in 150 healthy volunteers during development of entolimod as a radiation countermeasure and in a Phase 1 trial in a smaller number of cancer patients. While the immune response to entolimod does not compromise its utility for single-dose indications (e.g., protection against or mitigation of acute radiation syndrome), it effectively precludes numerous other potential applications requiring multiple doses over extended time periods. Such applications with proof of concept for TLR5 agonist therapeutic efficacy in preclinical studies include anticancer immunotherapy and prevention of metastasis[10,25–28] and protection of normal (non-tumor) tissues from the toxicity of anticancer radio/chemotherapy[6,14] and other types of stress and inflammatory syndromes[4,7,12]. Therefore, our key objective was to engineer a second-generation TLR5 agonistic entolimod derivative with substantially reduced neutralizing antigenicity and de novo immunogenicity.

To this end, we leveraged available structure–function knowledge of entolimod-TLR5 interactions and signaling[17] and state-of-the-art epitope prediction and mapping methodologies to complete a three-stage protein engineering program (Fig. 2). The obtained reengineered TLR5 agonist GP532 retains nearly full NF-κB signaling activity characteristic of entolimod (or flagellin) yet shows: (i) greatly enhanced resistance to neutralizing Abs; (ii) suppressed de novo immunogenicity; and (iii) completely eradicated inflammasome activation capability due to elimination of major B-cell epitopes, T-cell epitopes, and an inflammasome-activating domain, respectively.

Deimmunization is necessary for any nonhuman recombinant protein to be used for therapeutic administration[29] and is typically primarily focused on mapping and eliminating the T-cell epitopes responsible for the protein's de novo immunogenicity. To accomplish this for entolimod, we utilized a state-of-the-art epitope mapping and elimination methodology that is widely

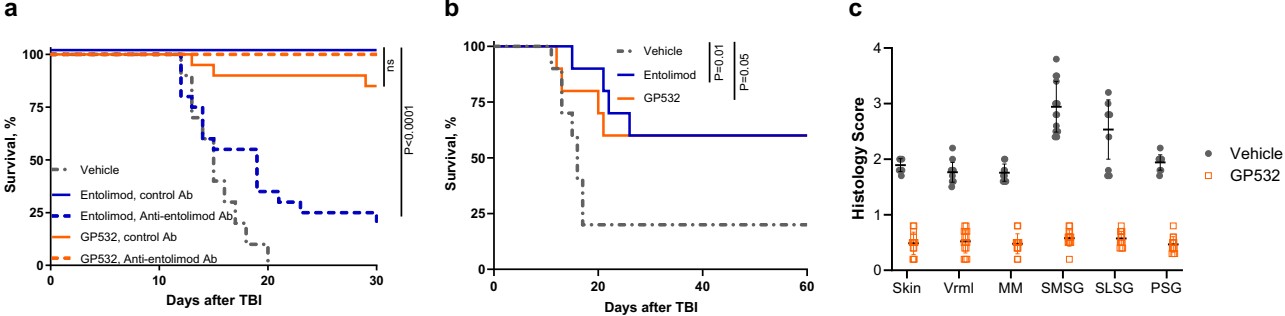

**Fig. 10 Therapeutic efficacy of entolimod and GP532 in mice exposed to lethal total body irradiation (TBI) or sublethal local H&N irradiation. a** Radioprotective potency of GP532 and entolimod administered to NIH Swiss mice (s.c., 1 μg/mouse) 30 min before lethal TBI (8.5 Gy) in the presence of "anti-entolimod Ab" (serum from entolimod clinical trial patients with high levels of entolimod-reactive neutralizing Abs) or "control Ab" (normal human serum with a very low level of entolimod-reactive Abs). $n = 10$ for vehicle-treated group, $n = 20$ for all other groups. **b** Radiomitigative activity of GP532 and entolimod administered to BALB/c mice (s.c., 1 μg/mouse, $n = 10$ per group) 24 h after lethal TBI (7.5 Gy). For **a**, **b**, P values by Log-rank test are shown; ns, not significant. **c** Tissue-protective activity of GP532 in female FVB/NJ mice ($n = 15$ per group) injected s.c. with 0.3 μg GP532 (or PBST as a negative control) 30 min before 15 Gy irradiation (X-ray) to the H&N region. H&E-stained sections of lips (skin, vermilion (Vrml), and mouth mucosa (MM)) and salivary glands (submandibular salivary glands (SMSG), sublingual salivary glands (SLSG), and parotid salivary glands (PSG)) collected 30 days post-irradiation were scored for injury on a scale of 0 (normal morphology) to 4 (most severe damage). Markers show individual sample scores ($n = 12-15$ independent samples (mice) per group); horizontal bars show the per group mean scores with error bars indicating SD. For all tissues, the difference in mean histology scores between PBST and GP532 groups was statistically significant ($P < 0.0001$ by two-tailed unpaired t test).

used for humanization of therapeutic antibodies[18]. Unlike B-cell epitopes, which are often topologically defined, T-cell epitopes are strictly linear, consisting of peptides generated by proteasomal degradation and presented by MHC molecules. Thus, we identified major T-cell epitopes in entolimod by a standard approach combining ex vivo screening of a synthetic peptide library with computational modeling. While generally straightforward, this approach has some challenges, including: (i) the substantial (hereditary) heterogeneity of MHC allotypes, and thus, T-cell epitopes, in the human population; and (ii) the fact that determination of the actual extent of suppression of de novo immunogenicity might only be accomplished through testing in humans. Nevertheless, we strongly believe that our demonstration of reduced de novo immunogenicity of GP532 vs. entolimod using ex vivo testing with representative PBMC samples indicates substantial success of this component of our deimmunization campaign and supports the likelihood that the introduced protein modifications will have a positive translational impact.

Elimination of major B-cell epitopes is a challenging task that is only rarely pursued in biological drug development (e.g., for immunotoxins[30]). However, this is of particular importance for flagellin-based TLR5 agonists due to the ubiquitous life-long exposure of humans to commensal and pathogenic enterobacteria abundantly expressing this bacterial motility protein. Unfavorable implications of this exposure for therapeutic use of flagellin derivatives are twofold: (i) the finding that a single injection of entolimod leads to a very rapid and robust neutralizing immune response in all tested human adults (as established in entolimod clinical trials) indicates that flagellin-specific immune memory cells are universally present in the human population and would presumably negate the therapeutic efficacy of flagellin-derived TLR5 agonists in multi-dose treatment regimens; and (ii) for ~10% of adult humans, a prohibitively high level of preexisting neutralizing flagellin-reactive Abs (also observed in entolimod clinical trials) would render them insufficiently responsive even to a single injection of entolimod. Such elevated Ab levels may reflect a recent (or present) bacterial infection, differences in response to commensal bacteria, etc.

Reducing neutralizing antigenicity is challenging due to the lack of a straightforward approach for identification and

elimination of B-cell epitopes. To overcome this, we applied a structure-guided[17] iterative approach, combining in silico prediction of candidate epitopes with site-directed mutagenesis and testing of mutant variants for resistance to signaling inhibition by neutralizing entolimod-reactive Abs. Available 3D structural information allowed us to tentatively predict topological epitopes and avoid mutations affecting critical interactions between the entolimod D1 domain and the TLR5 ectodomain. The lack of structural data for the D0 domain was partially mitigated by deletion of the entire N-terminal ND0 segment and partial truncation of CD0 at the C-terminus, which had only a minor effect on NF-κB activation. The latter truncation also eliminated both the third identified T-cell epitope and a domain required for NLRC4-mediated inflammasome activation[22]. Eliminating the possibility of inflammasome-mediated production of pro-inflammatory cytokines such as IL-1β and IL-18 is expected to only improve the safety of the drug.

GP532 also has improved pharmacological properties compared to entolimod, as illustrated by side-by-side PK/PD profiling in mice (Fig. 6a, b). Increased bioavailability of GP532 likely explains the extended duration of NF-κB signaling observed in mouse hepatocytes (Fig. 7).

Biocomparability of GP532 and entolimod was demonstrated by a series of comparative studies in mice. Both compounds induced very similar patterns of gene expression in mouse livers consistent with the previously defined mechanism of action of flagellin/entolimod involving activation of NF-κB and AP-1 pathways (Fig. 9). Most critically, GP532 demonstrated therapeutic efficacy comparable to entolimod in mouse models of lethal TBI (Fig. 10a, b) and sublethal local H&N irradiation (Fig. 10c). TLR5/NF-κB activation in mouse livers and radioprotection of lethally irradiated mice by GP532 were not affected by transfused human entolimod-reactive Abs, whereas such Abs dramatically suppressed entolimod's activity in the same assays (Figs. 8 and 10a).

These results set the stage for further clinical development of GP532 for therapeutic indications requiring extended, multi-dose treatment. While complete elimination of all neutralizing B-cell epitopes in a therapeutic protein without loss of its desirable activity is likely not feasible, we expect the level of

immunoresistance achieved in GP532 to strongly improve its clinical utility. At a minimum, most patients with prohibitively high levels of preexisting entolimod-reactive Abs should be responsive to at least short-term GP532 treatment; with a 30% inhibition threshold, the segment of the population rejected for treatment would be reduced from ~10% for entolimod to 1–2% for GP532 (see Supplementary Fig. 3b). At the same time, we expect that the reduced neutralizing antigenicity and de novo immunogenicity of GP532 will allow expanded treatment regimens of multiple doses administered over at least several weeks. Given the numerous potential clinical applications of TLR5 agonists, availability of a new drug such as GP532 that is not limited by immune inactivation is likely to have an important impact on human health.

## Methods

**Detection of entolimod-reactive Abs and entolimod-neutralizing activity in human serum samples**. Titers of entolimod-reactive Abs in human serum samples (e.g., samples collected from entolimod clinical trial subjects prior to entolimod dosing and at various time points post-dosing) were determined using an ELISA method in bridging format. Serum samples were collected from cancer patients dosed with entolimod (CBLB502) in trial no. NCT01527136 (ClinicalTrials.gov), which was performed at Roswell Park Cancer Institute (RPCI) under RPCI Study Number I196111 (Alex Adjei, MD, PhD, Principal Investigator). The protocol for this study was approved by the Institutional Review Board of RPCI and followed all relevant ethical regulations. Informed consent for the study, including collection of blood samples for measurement of anti-entolimod antibody levels, was obtained from all trial participants prior to their enrollment. Plates coated with entolimod were incubated with diluted serum samples and bound entolimod-reactive Abs were detected using biotin-labeled entolimod and streptavidin-poly-HRP conjugate. Neutralizing activity of serum samples positive for entolimod-reactive Abs was tested in a cell-based NF-κB-lacZ reporter assay (see below). Briefly, entolimod was preincubated with serum (or assay medium as a reference control) for 1 h before the mixture was added to NF-κB-lacZ reporter cells (HEK293-hTLR5::NF-κB-lacZ cells). Expression of the lacZ reporter gene was determined 20 h later by spectrophotometric measurement of β-galactosidase activity in cell lysates mixed with ONPG substrate (OD at 414 nm). Neutralization was assessed as the percent (%) inhibition of reporter activity in samples containing serum vs. medium (the reference control). During qualification of this assay prior to screening clinical samples, 30% inhibition was established as the cut-off indicating presence of entolimod-reactive neutralizing Abs in a serum sample.

**Cell-based NF-κB signaling activity assay**. NF-κB signaling activity of TLR5 agonizts including purified intermediate variants was measured using HEK293-hTLR5::NF-κB-lacZ reporter cells. This stable reporter cell line was developed at Cleveland BioLabs, Inc. (Buffalo, NY) by transducing human embryonic kidney cells (HEK293) stably expressing full length human TLR5 (Invitrogen, catalog number 293-htlr-5) with lentiviral vector Lenti-NFkB-LacZ containing an NF-κB-dependent reporter gene (lacZ, encoding β-galactosidase). The reporter cells were plated at a density of 30,000 cells/well in 96-well tissue culture-treated plates and incubated with serial dilutions of tested proteins at concentrations ranging from 0.0004 to 2.5 ng/mL for 16 to 20 h at 37 °C. After incubation, the cell lysis buffer containing the chromogenic substrate ortho-nitrophenyl-β-galactoside (ONPG) was added to the wells and β-galactosidase activity was measured as OD at 414 nm. EC50 values were determined from the dose-response curves using GraphPad Prism software.

**In vitro neutralization assay**. To quantify neutralizing antibody effects, NF-κB signaling activity induced by TLR5 agonists and intermediate variants in HEK293-hTLR5::NF-κB-lacZ reporter cells was measured in the presence vs. absence of human serum or entolimod-reactive Abs. For a standard neutralization assay, 50 µL of human serum was diluted 12.5-fold in assay medium and preincubated with 50 µL TLR5 agonist protein (2 ng/mL in assay medium) for 1 h at room temperature (RT) to allow antibody binding. As a reference control, 50 µL assay medium was used instead of serum. Next, 100 µL HEK293-hTLR5::NF-κB-lacZ cells (at 300,000 cells/mL) were added to each well, resulting in 50-fold dilution of serum and 0.5 ng/mL final concentration of TLR5 agonist protein. After 20 h incubation at 37 °C in 5% CO2, β-galactosidase activity was measured as described above. Neutralization activity was calculated as percent (%) inhibition, the percentage difference between the OD414 readings for reference control and serum samples. Neutralizing antibody titration assays were performed in the same way, except that the serum samples or entolimod-specific Abs were serially diluted before preincubation with TLR5 agonist proteins. Percent inhibition values were calculated for all dilutions and were plotted to generate neutralization titration curves. IC50 values (dilution at which % inhibition = 50) were determined from the titration curves using GraphPad Prism and were compared between proteins for each serum or antibody sample.

To map and monitor the iterative elimination of B-cell epitopes, we used human serum samples from 45 anonymous entolimod-naïve donors (selected for the highest neutralization activity from the total of 200 analyzed serum samples obtained from a commercial source, Bioreclamation IVT) and from subjects injected with entolimod in clinical trials (samples from patients P12 and P14 provided by CBLI). In addition, we used entolimod-reactive mouse monoclonal Abs (mAB4D11, mAb11D04, developed and kindly provided for this study by CBLI).

**T-cell epitope mapping**. T-cell epitope mapping was performed by Abzena (https://abzena.com) using the EpiScreen™ methodology[18] based on peptide library screening over a panel of 50 representative PBMC samples followed by in silico modeling of mapped epitopes to guide their elimination. This is described in detail in Supplementary Methods, Abzena Reports 1 and 2.

**Site-directed mutagenesis and protein engineering**. A DNA construct encoding the complete amino acid sequence of entolimod (Supplementary Fig. 1) inserted in a pET-49b(+)-based expression vector (EMD Bioscience) under control of the T7 promoter (as described in[2]) was provided by Cleveland BioLabs, Inc. (CBLI) and used as a starting point for engineering of all intermediate variants leading to GP532 (Supplementary Table 2). Site-directed mutagenesis (1–3 adjacent codons at a time) was performed using a standard primer extension approach as previously applied in[17]. Constructs with deletions and mutations assembled from different intermediate variants were generated using PCR and standard cloning techniques. All obtained constructs were verified by DNA sequencing and transformed into E. coli strain T7 Express Iq (New England BioLabs) for protein expression.

**Protein expression and purification**. Pilot expression of intermediate variants for initial testing was performed in 50 mL scale in LB/kanamycin media with IPTG induction. A small-scale protocol yielding up to 2 mg of >85% pure protein (depending on the variant) involved extraction by 2 M urea followed by mini-scale purification on Ni-NTA agarose (Qiagen). For more detailed characterization, key intermediate variants (33MX, 33ML) and GP532 were expressed in 1 L scale and purified by a previously described two-step protocol[17]: Ni-NTA gradient chromatography followed by gel filtration on Superdex G-200 (FPLC) and removal of residual endotoxin (Pierce detox resin); this yielded 20–30 mg of >90% pure protein. In a scaled-up GMP-compatible method developed for GP532 manufacturing, the second chromatographic step was replaced with more robust AEX on HiPrep Q FF 16/10 column (GE). This method was used to obtain ~200 mg of highly purified GP532 protein (>97% by SDS-PAGE, RP-HPLC, and SE-HPLC, see Supplementary Fig. 5) with fully removed endotoxin. Entolimod used in all experiments (provided by CBLI) was prepared by a similar method, as described in[2].

**In vitro immunogenicity testing**. The residual immunogenicity of GP532 in comparison with parental entolimod was tested by Abzena (https://abzena.com) using the EpiScreen™ DC:T-cell assay[20] with the same panel of PBMC from 50 healthy donors that was used for T-cell epitope mapping. This is described in detail in Supplementary Methods, Abzena Report 3.

**In vitro inflammasome activation assays**. An inflammasome test cell line, THP1-NLRC4 (InvivoGen) was used to evaluate inflammasome activation by TLR5 agonists and intermediate protein variants in the following assays:

*Measurement of IL-1β produced by activated THP1-NLRC4 cells using ELISA.* THP1-NLRC4 cells (100,000 cells/well) were incubated in 96-well plates with TLR5 agonist proteins in a range of concentrations from 0.005 to 20 ng/mL (0.13 to 800 pM) for 20 h at 37 °C, 5% CO2. IL-1β was measured in culture supernatants using the Human IL-1β/IL-1F2 DuoSet ELISA kit (R&D Systems). Briefly, Nunc Max-iSorp black immunoassay plates were coated with mouse anti-human IL-1β capture antibody and blocked with 1% BSA/PBS/0.05% Tween 20 assay buffer. Culture supernatants and recombinant human IL-1β standard solutions (50 µL/well, in duplicate) were incubated with the capture antibody for 2 h at RT followed by a 2 h RT incubation with biotinylated goat anti-human IL-1β detection antibody. The bound biotin-labeled detection antibody was detected with streptavidin-HRP (1 h at RT) and QuantaBlu fluorogenic peroxidase substrate (Thermo Fisher). Fluorescence (Relative Fluorescence Units, RFU) was measured after 30 min incubation at RT at 320 nm excitation and 420 nm emission wavelengths using a SpectraMax M3 plate reader. IL-1β concentrations were calculated from the standard curve using GraphPad Prism software and five-parameter logistic model.

*Concurrent detection of IL-1β produced by activated THP1-NLRC4 cells using HEK-Blue KD-TLR5 reporter cells.* In 96-well plates, THP1-NLRC4 cells (20,000 cells/well) were cocultured with HEK-Blue KD-TLR5 reporter cells (InvivoGen; 30,000 cells/well), in the presence of TLR5 agonist proteins at concentrations ranging from 0.04 to 385 pM for 20 h at 37 °C, 5% CO2. During incubation, IL-1β produced by THP1-NLRC4 cells as a result of inflammasome activation is secreted into the

culture medium where it reacts with HEK-Blue KD-TLR5 cells, inducing a secreted embryonic alkaline phosphatase (SEAP) reporter enzyme. SEAP activity was measured in culture supernatants in duplicate 25 µL aliquots using 100 µL Atto-Phos AP fluorescent substrate (Promega) at 435 nm excitation and 575 nm emission wavelengths.

*Detection of caspase-1 activity in THP1-NLRC4 cell cultures.* Using Corning 96-well white polystyrene microplates, THP1-NLRC4 cells (200,000 cells/well) were incubated with entolimod (18 to 142 pM) or GP532 (100 to 803 pM) for 5 h at 37 °C in 5% $CO_2$. Caspase-1 activity was measured by Caspase-Glo 1 Inflammasome Assay kit (Promega). Assay reagent (100 µL), containing Caspase-Glo 1 Buffer, Z-WEHD Substrate, and MG-132 Inhibitor was added to 100 µL cell culture and incubated at RT for 30 to 60 min. Luminescence (RLU) was measured using a SpectraMax M3 plate reader.

**Mouse studies.** Experiments performed in *Mus musculus* (laboratory mice) complied with all relevant ethical regulations for animal testing and research and were approved by the Institutional Animal Care and Use Committee (IACUC) of Roswell Park Cancer Institute (RPCI). C57BL/6, NIH Swiss, BALB/c, FVB/N, TLR5-Knockout (B6(Cg)-Tlr5tm1.2Gewr/J), and transgenic NF-κB-luciferase (Balb/C-Tg(IκBα-luc)Xen) reporter mice were used in this study; details on the source, sex and age of mice used in different experiments are provided in the corresponding Methods sections below and/or in figure legends.

**Pharmacokinetics/pharmacodynamics (PK/PD) profiling in mice.** PK/PD profiling of GP532 was performed in side-by-side comparison with entolimod following s.c. administration in C57BL/6 J mice (9 weeks of age, five female mice per time point). Both proteins were administered at a fixed dose (1 µg in 100 µL of PBS-0.05% Tween 80). One group of mice received vehicle alone to provide a baseline for analyses. Blood was collected by cardiac puncture at designated time points post-dose (15 min, 30 min, 1, 2, 4, 8, and 24 h). The levels of administered proteins and selected cytokine biomarkers including G-CSF (a marker of TLR5-NFκB signaling) and IL-18 (a circulating marker of inflammasome activation) in obtained serum samples were determined by corresponding ELISA protocols. Entolimod and GP532 concentrations in serum samples were measured following quantitative ELISA procedures previously developed for PK analysis of entolimod[3]. Briefly, Nunc MaxiSorp 96-well black immunoassay plates were coated with rabbit polyclonal anti-entolimod Abs that also bind GP532. After overnight incubation at 4 °C with appropriate dilutions of serum samples or respective calibration standards in a range of concentrations from 0.01 to 5 ng/mL (50 µL/well, in duplicate) and washing, the plates were incubated with biotinylated goat anti-entolimod detection Abs for 1 h at RT. After washing, the plates were incubated with Pierce Streptavidin-poly-HRP (Thermo Fisher Scientific) diluted 1:10,000 in Blocker Casein (Thermo Fisher Scientific) at RT for 1 h. Plates were then washed and incubated with QuantaBlu Fluorogenic Peroxidase Substrate (80 µL/well) at RT for 30-60 min. Fluorescence (RFU) was measured using a SpectraMax M3 plate reader (excitation at 320 nm, and emission at 420 nm). Protein concentrations were calculated from corresponding standard curves by GraphPad Prism using five-parameter logistic model and 1/response weighting. Concentrations of G-CSF and IL-18 in serum samples were measured using R&D Systems ELISA kits (Catalog #DY414 and 7625, respectively) according to the manufacturer's recommendations using equipment, development procedures, and statistical methods similar to those described above.

**Immunohistochemical analysis of NF-κB signaling in mouse livers.** C57BL/6 mice (females, 22–25 g body weight) were injected with entolimod or GP532 (s.c., 1 µg/mouse) and euthanized for preparation of liver sections at 30 min, 2 h or 24 h post-injection. 12 mm sections were prepared from fresh-frozen liver samples and fixed with 4% formaldehyde/PBS for 10 min. Sections were then washed in PBS and stained with rabbit monoclonal Ab against NF-κB p65 subunit (Cell Signaling, #8242, dilution 1:200) and rat monoclonal Ab against macrophage marker F4/80 (Bio-Rad, #MCA497GA, dilution 1:200) followed by secondary fluorochrome-conjugated donkey Abs from Jackson ImmunoResearch (anti-rabbit Cy3 and anti-rat AlexaFluor 488, dilution 1:500) and DAPI (blue) DNA stain. Stained sections were examined using a Zeiss AxioImager A1 microscope equipped with a X-100 epifluorescent light source. Images were captured with an AxioCam MRc digital camera and processed with AxioVision software (Carl Zeiss, Germany, rel4.8).

**Determination of TLR5 agonist activity and susceptibility to neutralization in transgenic NF-κB-luciferase reporter mice.** Transgenic NF-κB-luciferase reporter mice (Balb/C-Tg(IκBα-luc)Xen reporter mice, originally purchased from Xenogen and bred in our animal facility) were 14–23 weeks of age at the time of this experiment. Groups of 3–4 mice each were injected i.v. with 100 µl of PBS or 1:10 diluted human serum. Serum samples that were used included those from entolimod clinical trial subjects (p12 and p14, pooled with high levels of entolimod-reactive neutralizing Abs, referred to as anti-entolimod Ab) and normal human serum samples with very low levels of entolimod-reactive neutralizing Abs (referred to as isotype control Ab). 2.5 h after serum transfer, mice were injected s.c. with PBS, 1 µg entolimod or 1 µg of GP532 (100 µl volume). Three hours later,

mice were injected i.p. with 150 mg/kg luciferin (200 µl injection volume per 25 g mouse body weight). Mice were imaged using an IVIS Spectrum Imaging System within 30 min of luciferin injection. Five images spaced 2 min apart were collected per mouse (from 12 to 20 min post luciferin administration). Bioluminescence (total flux) was quantified using Living Image software. Group 1 (vehicle treatment) had two males and two females, Group 2 (entolimod + isotype control Ab) had two females and one male, Group 3 (entolimod + anti-entolimod Ab) had two males and two females, Group 4 (GP532 + isotype control Ab) had four females, and Group 5 (GP532 + anti-entolimod Ab) had four males.

**RNAseq-based comparative transcriptomics in mouse livers.** Pathogen-free C57BL/6 male mice were obtained from The Jackson Laboratory and sex and age-matched TLR5-KO (B6(Cg)-Tlr5tm1.2Gewr/J) were obtained from our colony at RPCI. Mice were housed in microisolator cages in a laminar flow unit under ambient light. At 12 weeks of age, groups of mice (n = 3) received a single s.c. injection of GP532 (0.3 µg), entolimod (0.3 µg), or vehicle (PBST). Livers were collected at 30 min or 24 h post-treatment and flash frozen. Total and small RNA was extracted using the miRNeasy mini kit (Qiagen) following the manufacturer's recommended protocol. DNAse digestion was performed to remove any residual genomic DNA followed by additional washes prior to elution of high-quality RNA in RNase-free water. The purified RNA was quantitatively assessed using a Qubit Broad Range RNA kit (Thermo Fisher) and its concentration was determined by Ribogreen fluorescent binding. The RNA was further evaluated qualitatively using RNA Nanotape on the 4200 Tapestation (Agilent technologies), where sizing of the RNA was determined, and a qualitative numerical score (RINe) was assigned. Sequencing libraries were prepared with the RNA HyperPrep Kit with RiboErase (HMR) kit (Roche Sequencing Solutions) from 500 ng total RNA following the manufacturer's recommended protocol. The resulting pool of DNA libraries was then loaded into the appropriate NovaSeq Reagent cartridge for 100 cycle paired end sequencing and sequenced on a NovaSeq6000 following the manufacturer's recommended protocol (Illumina, Inc.). The quality of the sequencing data quality was assessed via FastQC (http://www.bioinformatics.babraham.ac.uk/projects/fastqc). Reads were aligned to the mouse reference genome (UCSC mm10/GRCm38) with STAR RNA-seq aligner[31] using annotation from the same source. Reads were counted using featureCounts[32] using the same annotation. Differential gene expression and normalized counts were calculated using DESeq2[33]. Further DE comparison, analysis and visualization were performed using both Python and R programming languages. KEGG enrichment was performed with R clusterProfiler package[34].

**Radioprotection in lethally irradiated mice.** Efficacy of entolimod and GP532 in protecting mice against lethal acute radiation syndrome following high dose TBI and the effect of neutralizing Abs on this activity was tested as follows. A control group of female NIH Swiss mice (12 weeks old, n = 20) was injected (s.c., 0.1 mL) with PBS-0.1% Tween 80 (PBST) vehicle 30 min before TBI. Two additional groups of age-matched NIH Swiss mice (n = 40/group) were injected with human serum (1:10 dilution, i.p., 0.2 mL), with one group receiving serum from entolimod clinical trial patients with high levels of entolimod-reactive neutralizing Abs and the other group receiving normal human sera with a very low level of entolimod-reactive Abs. One hour later, 20 mice from each serum-injected group was injected with entolimod and 20 mice were injected with GP532 (both administered s.c., 1 µg per mouse); TBI was applied to these mice 30 min later. All mice received a single lethal dose of TBI (8.5 Gy) using an irradiator with a Cesium-137 source (2200 Ci) and a rotating platform to ensure even dose delivery to all tissues (J.L. Shepherd and Associates). Survival was monitored for 30 days post-TBI.

**Radiomitigation in lethally irradiated mice.** Efficacy of entolimod and GP532 in mitigating lethal acute radiation syndrome when administered after high dose TBI was tested as follows. Pathogen-free BALB/c mice (13 weeks old) were obtained from The Jackson Laboratory and housed in microisolator cages in a laminar flow unit under ambient light. Mice were exposed to 7.5 Gy TBI as described above for the radioprotection model. At 24 h post-TBI, groups of irradiated mice (n = 10/group, all females) were injected s.c. with PBST vehicle or 1 µg entolimod or GP532. Survival was monitored for 60 days post-TBI.

**Protection against tissue damage caused by sublethal head and neck (H&N) irradiation in mice.** The ability of GP532 to protect salivary glands and mouth mucosa tissues from localized sublethal H&N irradiation was determined using female FVB/NJ mice (n = 15/group, 10–12 weeks old, Jackson Laboratories). GP532 (0.3 µg) or PBST vehicle was administered by s.c. injection 30 min prior to localized irradiation (15 Gy) to the H&N area. Irradiation was performed with a Philips RT250 250 kV X-ray irradiator and lead shielding to protect the other regions of animal's body. Thirty days post-irradiation, mice were euthanized, and tissues were collected and fixed in formalin. Tissue sections were stained with hematoxylin and eosin for histological examination. The components of the lips (skin, vermilion, mouth mucosa) and salivary glands (submandibular containing mixed cells; sublingual gland containing mucin acini; parotid gland containing serous acini) were evaluated for radiation-induced morphological alterations by a qualified pathologist, who was blinded with respect to the treatment groups.

Radiation-induced injury was scored semi-quantitatively using a scale from 0 (normal morphology) to 4 (greatest degree of injury).

**Statistics and Reproducibility**. Numbers of animals, numbers of replicates in assays, and *P* values are provided in figure legends. Mean data are presented with errors bars indicating standard deviation (SD). Statistical tests were performed using GraphPad Prism software. The effects of entolimod-neutralizing antisera on the NF-κB signaling activity of different TLR5 agonists and intermediate variants were compared by ANOVA (e.g., Fig. 3b). Radio-protective and -mitigative effects of TLR5 agonists in the presence or absence of entolimod-neutralizing Abs (Fig. 10a, b) were assessed by comparing mouse survival in different treatment groups by Log-rank test. Scores for radiation-induced tissue damage (Fig. 10c) were compared between groups by two-tailed unpaired *t* test. For all statistical tests, *P* values ≤0.05 were considered statistically significant.

**Reporting Summary**. Further information on research design is available in the Nature Research Reporting Summary linked to this article.

## Data availability
The authors declare that all data supporting this study are available within the article and its supplementary information files, may be obtained from the corresponding author upon reasonable request, or in the case of our RNAseq experiment comparing gene expression in livers of wild type and TLR5$^{-/-}$ C57BL/6 mice treated with vehicle, entolimod, or GP532, are available in NCBI's Gene Expression Omnibus (GEO) database under accession number GSE163748. Source data for graphs presented in the article (Figs. 1, 3b, 5, 6, 9, and 10) are provided in the Supplementary Data file.

## Material availability
Biological materials specific to this study (entolimod and its derivatives described in this study, including GP532) are available from the authors for research purposes only upon request and with a Material Transfer Agreement.

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

## Acknowledgements
This work was supported by contracts from Cleveland Biolabs, Inc. to L.G.B and A.V.G., Genome Protection, Inc. to C.M.B. and A.V.G., and National Cancer Institute (NCI) grant P30CA016056, involving the use of Roswell Park Comprehensive Cancer Center's Shared Resources, to A.V.G. We thank Kellee Greene and Alyssa Aldrich for excellent technical assistance and Patricia Stanhope Baker for help with manuscript preparation.

## Author contributions
V.M. contributed to study planning, designed the study, analyzed data, and wrote the manuscript; O.V.K., I.A.B., and C.A.B. conducted experiments; I.M. analyzed data and provided bioinformatics support; C.M.B., L.G.B., and A.S.G. conducted experiments and analyzed data; A.A.P. analyzed data and wrote the manuscript; I.A.T. analyzed data; Y.N.K. designed the study and analyzed data; E.L.A. contributed to study planning and analyzed data; A.V.G. and A.L.O. contributed to study planning, designed the study, analyzed data, and wrote the manuscript.

## Competing interests
The authors declare the following competing interests: A.V.G. is a shareholder of and consultant for Cleveland BioLabs, Inc. (CBLI) and Genome Protection, Inc. (GPI), companies that contributed to the development of entolimod and GP532. Research in Dr. Gudkov's laboratory is supported by funding from both companies. A.L.O. is a consultant for GPI and the work of his laboratory on this project was supported by a research contract from Buffalo BioLabs. C.M.B. and L.G.B. have research contracts from GPI and CBLI, respectively. A.A.P., Y.N.K., and E.L.A. are consultants of GPI. All other authors declare no competing interests.
