## [Peer Review File · Communications Biology]

Reviewers' comments:

Reviewer #1 (Remarks to the Author):

In this very thorough and comprehensive paper Meet and coworkers describe a "deimmunized" and pharmacologically optimized entolimod variant, GP532 and present extensive preclinical data. I should note that I am very skeptical of B cell epitope removal as a means of reducing recognition by circulating antibodies, which constitutes an important part of the protein engineering strategy. Nonetheless B cell epitope de-immunization is widely pursued and therefore any concerns about its utility should not be held against the authors. In fact they should be commended for a very careful study. Hence I recommend publication in *Communication Biology* with enthusiasm. The authors should address (optional) the following questions which could further strengthen the paper.

- 1) Information on stability of GP532 (at least T_m for example) should be provided. Also is the protein more stable in serum which could explain its better PK?
- 2) The authors show in Fig 6 that GP532 has improved PK in mice. But why? There is no data showing lower recognition by mouse Ig. Could it be that GP532 is simply more stable and that is what is responsible for the better PK? That should be explicitly stated.
- 3) There is no analysis of the mouse PK data. C_{max} is higher for GP532 but why? A better analysis of PK would be helpful. Along these lines the authors want to comment on whether the higher biological activity of GP532 is directly a consequence of the greater AUC.

Reviewer #2 (Remarks to the Author):

In their manuscript: "A deimmunized and pharmacologically optimized Toll-like receptor 5 agonist for therapeutic applications" the authors report on the de-immunization of Entolimod. Entolimod, a pharmacological derivative of the natural Toll-like receptor 5 (TLR5) agonist flagellin, has strong therapeutic potential for several indications including radioprotection and cancer immunotherapy. However, upon administration, entolimod induced a rapid neutralizing immune response, presumably due to flagellin-reactive immune memory derived from life-long exposure of adult humans to flagellated enterobacteria. To reduce the antigenicity and immunogenicity of Entolimod, the authors used structure-guided reengineering to develop a new, substantially deimmunized entolimod variant they named GP532.

The experiments were carried out in several stages: First they identified the minimal functional core of entolimod. Next, they identified and removed B cell epitopes (yielding mutant 33MX). Next, they identified and removed T cell epitopes. Next, they eliminated the inflammasome-activating domain (actually it was achieved by the C-terminal deletion that removed T cell epitope 3). Finally, and along the stages of the process, they carried out functional assays for functionality, antigenicity and immunogenicity.

They show that GP532 induces TLR5-dependent NF- κ B activation like entolimod, but is smaller and has mutations that eliminate an inflammasome-activating domain, major neutralizing B-cell epitopes, and common T-cell epitopes. They further show that GP532 is resistant to human entolimod-neutralizing antibodies in cell and animal models and shows reduced de novo immunogenicity *ex vivo*. The authors claim that GP532 also has improved bioavailability, a stronger pharmacodynamic effect on key cytokine biomarkers, and a longer-lasting effect on NF- κ B signaling. Like entolimod, GP532 demonstrated potent prophylactic and therapeutic efficacy that was shown in mouse models of lethal acute radiation syndrome and local head-and-neck irradiation mimicking anticancer radiotherapy.

The authors describe that "Taken together, the results of their comparative analysis of entolimod and GP532 confirm that GP532 retains comparable NF- κ B-stimulating activity while being improved in terms of PK, PD, duration of NF- κ B response, lack of inflammasome

activation and resistance to neutralizing antibodies both in vitro and in vivo". The authors conclude that these results establish GP532 as a highly optimized TLR5 agonist suitable for multi-dose therapeutic regimens and for patients with high titers of preexisting flagellin-neutralizing antibodies.

This study is a tour-de-force study, obviously very well designed and carried out. The manuscript is very well presented, including the incredibly long supplement with 10s of pages of the contractor Abzena reports. I have a few questions the authors can relate to in what I consider a very minor revision:

- 1) The authors and the literature suggest that most people have pre-existing antibodies to entolimod, which is why most patients make anti-drug antibodies (ADA) against it. Still, to establish that most of the tested individuals have a recall immune response it would have more useful if they tested individually IgG, IgM and in fact IgA (that should be common in the response to gut microbiome and its components). Instead, in Figure 1 and Table S1 they show total antibodies. Why?
- 2) The final version, GP532, has a C-terminal end comprising a thrombin cleavage site followed by a His tag. I do not know that a His Tag is an accepted part of a biopharmaceutical. I do wish the authors provide evidence that it is (or describe its removal during the purification process of GP532, which is not a part of the purification process presented here).
- 3) ONPG hydrolysis is routinely measured at 420 nm. Why did the authors read the reactions at 414 nm?

Response to Reviewers

(Mett, V., et al. *A deimmunized and pharmacologically optimized Toll-like receptor 5 agonist for therapeutic applications.*)

Below we provide responses to all of the reviewers' critical comments and note any related revisions made to the manuscript.

In addition, please note that a number of changes were made to the manuscript in order to better comply with the journal's formatting guidelines for the number of display items and length of text. Only key changes are tracked in the resubmitted files (as noted in the list below); however, we are happy to provide versions with *all* changes tracked upon request.

Changes to the manuscript include:

- Reduction of the length of the Abstract from 195 words to 154 words. (not tracked)
- Addition of text in the Results section to address issues raised by the reviewers (additions in red font beginning on Line 308 and Line 315 of the revised manuscript).
- Reduction of the number of display items from 13 to 10 by moving Tables 1 and 2 to the Supplementary Information (new Tables S3 and S4) and by eliminating Figure 11. A reformatted version of Figure 11 has been added to Figure 10 as panel C so that all information on the therapeutic efficacy of GP532 in mouse irradiation models is in a single figure. The displayed data in the revised manuscript is identical to that in the original manuscript, only its location (and formatting for Fig. 11) was changed. These changes are not tracked in the resubmitted article document, but the original and revised versions of Figure 10/11 are attached to this response (**Attachment 1**).
- Renumbering of Supplementary Tables numbered S3 to S6 in the original manuscript to be S5 to S8 in the revised manuscript. (not tracked)
- Numerous small edits made throughout the text in order to reduce its size. No content was eliminated, it was just made more concise. (not tracked)
- Relocation of several larger pieces of text describing Supplementary Table S6 (new numbering) and Supplementary Figures S8, S9 and S11 from the Results section to corresponding places in the Supplementary Information file. These deletions are indicated by strikethrough in the revised manuscript; see Lines 232, 378, 386 and 394 for text related to Table S6, Figure S8, Figure S9, and Figure S11, respectively.
- Revision of the Data Availability section to indicate that our RNA-seq gene expression profiling data has been deposited in NCBI's Gene Expression Omnibus (GEO) database and provide the corresponding accession number. (red font, starting on Line 766)
- Addition of a statement on Availability of Biological Materials as requested in the journal's Editorial Policy Checklist. (red font, starting on Line 770)
- Addition of the ClinicalTrials.gov registration number for clinical trials from which human serum samples were obtained for our study, as requested in the journal's Editorial Policy Checklist. (red font, starting on Line 100)

(Line numbers reflect those in the file following conversion to pdf format by the journal's submission portal.)

Reviewer #1 (Remarks to the Author):

1A) Information on stability of GP532 (at least T_m for example) should be provided. Also is the protein more stable in serum which could explain its better PK?

1B) The authors show in Fig 6 that GP532 has improved PK in mice. But why? There is no data showing lower recognition by mouse Ig. Could it be that GP532 is simply more stable and that is what is responsible for the better PK? That should be explicitly stated.

RESPONSE: To address these questions and in preparation for GMP manufacturing, we have performed stability studies for GP532. For these studies, we used GP532 lot “RS3”, which was produced using our finalized industrial expression-purification method (illustrated in **Supplementary Figure S5** of the manuscript) and is described by the formal **Certificate of Analysis** attached to this response (**Attachment 2**). The obtained results were compared with similar studies performed for GMP-manufactured entolimod. Briefly, accelerated stability studies of GP532 monitored by RP-HLC (**Supporting Figure 1A** below) showed relatively slow degradation of the protein (via aggregation and precipitation) during storage at +4°C (over 80% of the initial amount of GP532 remained intact after one month of storage). At room temperature (+20°C), degradation of the protein is notably accelerated, but the protein remains reasonably stable for at least 3 days. At an even higher temperature (37°C), GP532 remains fully stable for at least more than one day (data not shown), which is sufficient with respect to in-use stability. It is important to emphasize that stability of GP532 is strongly dependent on the protein concentration and this is substantially (orders of magnitude) lower in biological *in vitro* or *in vivo* assays (typical dilutions $\leq 1 \mu\text{g/mL}$) than in stability studies performed with a stock solution (concentration = 0.6 mg/mL in the studies shown here). While it is not feasible to use RP-HPLC to monitor protein stability in the context of biological assays with low protein concentrations, in-use stability of GP532 (comparable to that of entolimod) is supported by the results of bioassays involving preincubation of the protein with serum-containing media.

That said, we have no indications that GP532 is more stable than entolimod (as a potential explanation for our observation of the improved PK profile of GP532 vs entolimod). Rather, at least in the form of a relatively concentrated drug product, GP532 shows somewhat lower long-term stability than entolimod as illustrated in **Supporting Figure 1B**. It should be noted that this comparison is not perfect since the entolimod used is a GMP-manufactured drug with fully optimized formulation, while GP532 formulation optimization is still ongoing and we expect to ultimately reach the same or better stability for this drug. Upon completing formulation optimization, extensive stability studies including (but not limited to) serum stability as proposed by Reviewer #1, will be conducted as an essential component of regular pre-IND activity.

Finally, based on the Reviewer’s request for a comparison of melting temperature (T_m) values for entolimod and GP532, we performed Differential Scanning Calorimetry (DSC) for both proteins. However, as shown in **Supporting Figure 1C** below, the DSC analysis failed to identify a conventional phase transition that could be characterized by a definitive T_m for either entolimod and GP532. This observation may have an interpretation beyond technical problems. Indeed, the D0 domain of flagellin (which is the same as in entolimod) is likely to be somewhat disordered and, therefore, typically not discernable by X-ray crystallography (as was the case in our previous structure-function studies [ref. 17 in the manuscript]).

In summary, at this point, we do not have data to support the interpretation that the improved PK profile of GP532 compared to entolimod is driven by a difference in stability. We hypothesize that the improved PK profile may at least partially result from the reduced size of GP532, which may facilitate its faster delivery into circulation from the point of s.c. injection. In addition, GP532 is more hydrophobic than entolimod (having many Asp residues replaced by Ala). This might be another feature (beyond size) that contributes to sustainability of the protein in circulation by, for example, reducing access for proteases and/or complex formation with other blood constituents.)

To acknowledge this issue and provide at least a hypothetical interpretation, we have added the following text to the Results section of the revised manuscript (added text indicated by red font beginning at Line 303): “The exact reason(s) for the improved PK profile of GP532 vs entolimod remains unknown. Comparative stability studies did not show increased stability of GP532 (data not shown). Among possible

explanations, the smaller size (by ~ 30%) and higher hydrophobicity (due to replacement of many Asp residues with Ala) of GP532 may contribute to its greater bioavailability and/or sustainability in circulation.”

Supporting Figure 1A. Stability profile of GP532 (lot RS3, 0.6 mg/mL in PBS) at a range of storage temperatures (from -80°C to +20°C), as monitored by a developed RP-HPLC-based assay. Average measurements for duplicate samples are shown (SD values were $\leq 2\%$ in all cases and are not indicated here), with the amount of protein remaining at a given time point expressed as a percentage of that at the start of the experiment. Measurements for samples stored at +4°C and -20°C started 3 days and 1 week into experiment, respectively.

Supporting Figure 1B. Comparison of the stability of GP532 (lot RS3) and two GMP-manufactured lots of entolimod drug product (lots 16COA01 and 11COA01) upon storage at 4°C for up to 12 weeks. The amount of intact protein remaining at 2, 4 and 12 weeks was monitored by RP-HPLC and is shown on the graph as a percentage of the initial amount.

Supporting Figure 1C. Differential Scanning Calorimetry (DSC) assay results for entolimod (labeled 502, left panel) and GP532 (labeled 532, right panel).

2) There is no analysis of the mouse PK data. Cmax is higher for GP532 but why? A better analysis of PK would be helpful. Along these lines the authors want to comment on whether the higher biological activity of GP532 is directly a consequence of the greater AUC.

RESPONSE: Although this comment is generally related to the topic discussed above (and thus is subject to some aspects of our response to it), here, we provide a separate response focusing on the Cmax and AUC PK parameters and mechanism of action of GP532 relative to entolimod.

In response to the Reviewer’s request for a more detailed analysis of PK data, we provide values of the three standard parameters, Tmax (peak time), Cmax and AUC, computed for the same curves as shown in **Figure 6A** of the manuscript (**Supporting Table 1**). Values for both Cmax and AUC were comparably higher for GP532 vs entolimod, with the effect (indicated by the ratio of GP532/entolimod) appearing to be stronger at the level of AUC (3.1 for AUC vs 2.1 for Cmax).

PD analysis of the biomarker cytokine G-CSF, which was proven mechanistically relevant by previous studies with entolimod [ref. 5 in the manuscript], provides a proxy for the biological activity of TLR5 agonists. Data for the Tmax, Cmax, and AUC of G-CSF following administration of GP532 or entolimod in mice in the experiment shown in **Figure 6B** of the manuscript are listed below in **Supporting Table 1**. Consistent with our PK analysis, values for G-CSF Cmax and AUC parameters were higher for GP532 than for entolimod, although the differences were more subtle.

These findings generally support the Reviewer’s conjecture that “higher biological activity of GP532 is directly a consequence of the greater AUC”. However, from these data alone, we cannot unequivocally conclude whether the increased activity of GP532 is driven primarily by AUC or Cmax.

As suggested by the Reviewer, we have modified the Results section of the manuscript to describe the likely link between PK and bioactivity as follows (see added text in red font starting on Line 309): “The observed increase in AUC for G-CSF (2.3-fold for GP532 vs entolimod) likely reflects the comparable increase in PK parameters mentioned above.”

Supporting Table 1. Additional analysis of PK and PD data

	entolimod	GP532	Ratio GP532/ entolimod
Pharmacokinetics (Figure 6A)			
Tmax, hr	0.5	0.5	1
Cmax, ng/mL	4.2	9.0	2.1
AUC, ng*hr/l	6.2	19.2	3.1
Pharmacodynamics, G-CSF in serum (Figure 6B)			
Tmax, hr	2	4	2
Cmax, pg/ml	15,124	19,733	1.3
AUC, pg*hr/l	91,576	211,294	2.3

Reviewer #2 (Remarks to the Author):

1) The authors and the literature suggest that most people have pre-existing antibodies to entolimod, which is why most patients make anti-drug antibodies (ADA) against it. Still, to establish that most of the tested individuals have a recall immune response it would have more useful if they tested individually IgG, IgM and in fact IgA (that should be common in the response to gut microbiome and its components). Instead, in Figure 1 and Table S1 they show total antibodies. Why?

RESPONSE: We understand and agree with this opinion, and we would be happy to provide antibody subtype data if they were available. However, the data on human antibody responses following entolimod administration that are included in this paper as the rationale for development of a next-generation deimmunized TLR5 agonist were obtained in clinical trials by CBLI and kindly shared with us. Unfortunately, at the time of data collection, no detailed measurements to differentiate between different classes of immunoglobulins (IgG, IgM and IgA) were conducted. Appreciating the importance of such measurements, we are committed to including them in future clinical trials of GP532 expected to be conducted by Genome Protection, Inc.

2) The final version, GP532, has a C-terminal end comprising a thrombin cleavage site followed by a His tag. I do not know that a His Tag is an accepted part of a biopharmaceutical. I do wish the authors provide evidence that it is (or describe its removal during the purification process of GP532, which is not a part of the purification process presented here).

RESPONSE: The question of the acceptability of a His affinity tag (or any artificial sequence) in therapeutic proteins has been a subject of regulatory scrutiny and numerous debates. To our best knowledge, the current policy of the US FDA with respect to such artificial sequences depends, to a large extent, on the origin of a given protein. Thus, affinity tags are generally discouraged for biologics based on/derived from endogenous human proteins (although they may be deemed acceptable if compelling justification and proof of safety and low immunogenicity is provided). In contrast, no special restrictions on affinity tags are typically applied to heterologous (e.g., bacterial) proteins, whether natural or reengineered (as in case of GP532). From a regulatory perspective, the His tag on GP532 is just as foreign to a human organism as other segments of the protein, especially those modified by mutagenesis of the parental flagellin-derived sequence. This notion is supported by the existence of numerous artificial sequence-containing biologics approved by the FDA for clinical studies, including the parent of GP532, entolimod. Entolimod contains a His tag and streptokinase cleavage site at its N-terminus (see **Supplementary Figure S1** in the manuscript). In engineering GP532, we minimized and moved the His tag to the C-terminus of the protein for two reasons:

(i) It allows more efficient purification due to rapid removal of abortive translation products using an IMAC column. Such abortive translation products are commonly formed in the course of protein overexpression, but in this case would not include the C-terminal His tag and thus would be eliminated during IMAC purification.

(ii) More importantly, we found that deleting the 13 C-terminal residues of entolimod for the purpose of eliminating inflammasome activity also negatively impacted its desirable NF- κ B signaling activity, but that this could be fully prevented by replacing the deleted segment with a His tag.

Therefore, presence of a C-terminal His tag in GP532 is justified not only by the purification advantage it provides, but also by its necessity for preservation of the primary bioactivity of the protein. The latter argument typically provides sufficient justification for FDA acceptance of such a sequence.

Nevertheless, an argument against a particular sequence tag could still be made if it was found to be strongly immunogenic. However, despite some early concerns, the His6 tag has not been identified as a strong immunogen by any studies. Moreover, in the course of engineering GP532: (i) we have not observed any impact of His tag presence/absence or location (N- vs C-terminus) on neutralization of signaling activity by ADAs, and (ii) analysis of the intermediate entolimod variant 33MX, which has the same C-terminal tag as GP532, did not identify any prominent T cell epitopes in that part of the sequence (as reflected in the corresponding report from Abzena (report 1) provided in the Supplementary Information).

In summary, based on the above considerations and our previous extensive communications with the FDA about entolimod, we do not anticipate any regulatory difficulties due to the presence of a His tag in GP532.

3) ONPG hydrolysis is routinely measured at 420 nm. Why did the authors read the reactions at 414 nm?

RESPONSE: In short, the use of this particular wavelength was dictated by the equipment available for our use at the time of assay development. Our assay protocol was initially developed using a Labsystems Multiskan plate reader equipped with a 414 nm optical filter. For consistency, this wavelength setting was kept the same after switching to our current plate reader, a SpectraMax M3 instrument. Since the ONPG hydrolysis product (ONP) has a broad absorption spectrum with a maximum between 405 and 420 nm (e.g., see: https://www.researchgate.net/figure/Comparison-of-UV-spectra-of-ONPG-ONP-and-galactoside-100-mM-and-b-galactosidase-1-mM_fig5_50989259), readings obtained at 414 nm are expected to be close (or slightly higher than) those obtained at 420 nm. The 414 nm setting was used consistently for all of our samples.

Thus, while we understand the Reviewer's comment that 420 nm is a more typical setting, we are confident that this technical difference did not affect our results in any way.

ORIGINAL Figures 10 and 11

Figure 10. Comparison of the radio-protective and -mitigative efficacy of entolimod and GP532 and its sensitivity to neutralization in mouse models of lethal total body irradiation (TBI). **A.** Radioprotective potency of GP532 vs. entolimod administered (1 $\mu\text{g}/\text{mouse}$, s.c.) 30 min prior to lethal TBI (8.5 Gy) of NIH Swiss mice ($n=20$ per group) in the presence of “anti-entolimod Ab” (serum from entolimod clinical trial patients with high levels of entolimod-reactive neutralizing Abs) or “control Ab” (normal human serum with a very low level of entolimod-reactive Abs). **B.** Radiomitigative activity of GP532 and entolimod administered (1 $\mu\text{g}/\text{mouse}$, s.c.) 24 hrs after lethal TBI (7.5 Gy) of BALB/c mice ($n=10$ per group). P-values by Log-rank test are shown; ns, not significant.

Figure 11. GP532 protects mouse mucosal tissues and salivary glands against localized radiation to the head and neck (H&N) area. Female FVB/NJ mice ($n = 15$ per group) were injected s.c. with 0.3 μg GP532 (or PBST as a negative control) 30 min prior to receiving 15 Gy irradiation (X-ray) to the H&N region. Sections of tissues collected 30 days post-irradiation were stained with hematoxylin and eosin and scored for radiation injury on a scale of 0 (normal morphology) to 4 (most severe damage). Data are presented as the mean \pm SD. * indicates $P < 0.0001$ by unpaired t-test (two-tailed). MM, mouth mucosa; SMSG, submandibular salivary glands; SLSG, sublingual salivary glands; PSG, parotid salivary glands.

REVISED Figure 10 (Figure 11 eliminated)

Figure 10. Therapeutic efficacy of entolimod and GP532 in mice exposed to lethal total body irradiation (TBI) or sublethal local H&N irradiation. **A.** Radioprotective potency of GP532 and entolimod administered to NIH Swiss mice (s.c., 1 $\mu\text{g}/\text{mouse}$, $n=20$ per group) 30 min before lethal TBI (8.5 Gy) in the presence of “anti-entolimod Ab” (serum from entolimod clinical trial patients with high levels of entolimod-reactive neutralizing Abs) or “control Ab” (normal human serum with a very low level of entolimod-reactive Abs). **B.** Radiomitigative activity of GP532 and entolimod administered to BALB/c mice (s.c., 1 $\mu\text{g}/\text{mouse}$, $n=10$ per group) 24 hrs after lethal TBI (7.5 Gy). For **A-B**, P-values by Log-rank test are shown; ns, not significant. **C.** Tissue-protective activity of GP532 in female FVB/NJ mice ($n=15$ per group) injected s.c. with 0.3 μg GP532 (or PBST as a negative control) 30 min before 15 Gy irradiation (X-ray) to the H&N region. H&E-stained sections of lips (skin, vermilion (Vrml), and mouth mucosa (MM)) and salivary glands (submandibular salivary glands (SMSG), sublingual salivary glands (SLSG), and parotid salivary glands (PSG)) collected 30 days post-irradiation were scored for injury on a scale of 0 (normal morphology) to 4 (most severe damage). Mean \pm SD is shown. For all tissues, the difference in mean histology scores between PBST and GP532 groups was statistically significant ($P<0.0001$ by two-tailed unpaired t-test).

Attachment 2: Certificate of Analysis for GP532 lot RS3

CERTIFICATE OF ANALYSIS GP532 Reference Standard RS-3 Lot 6-20-1x

Prepared by	Genome Protection, Inc., 640 Ellicott Street, Suite 444, Buffalo, NY 14203
Expressed and purified from E. coli strain	KFJq/pET-49h(+)-GP532
Initial lot size	100 vials
Date of manufacturing	10 June 2019
Vial label	RS-3 6.20 (1x) ¹
Volume per tube	0.5 mL
Storage buffer	1x PBS ²
Storage conditions	-70°C ±10°C
Concentration	0.84 mg/mL ³
Purity by SDS PAGE ⁴	100% (Figure 1)
Purity by RP HPLC	100% (Figure 2)
Purity by SE HPLC	100% (Figure 3)
GP532 sequence ⁵	MSGLRINSKDDAAGQAAANRATSNIKGLTQASRNAA DGSLAQITTEGALNEINNLQRVRELSVQATAGANADA ALKAIQAEIQRLLEIDRVSQQTQAAA VKVLSQDNAMA IQVGANDGAAITIDLQKIDVKSLGLDGFVNNSPGSTANP LASIDSALSKVDVRSRLGAIQNRFDSAITNLGNTVTNL NSARSRIEDADYATEVSQMSKAQILDQAGTSTLAQLVP RGSHHHHHHG
Molecular mass ⁶	24.9 kDa
pI ⁶	5.8 kDa
Molar extinction at 280 nm ⁷	1490 [Abs of 1 mg/mL solution = 0.06]
Biopotency	EC ₅₀ 0.117 ng/mL (HEK293-TLR5-Luciferase reporter assay)
Inflammasome-directed signaling activity	Below LOD (IL-1β induction in THP1-NLRC4 cells)
Endotoxin	0.79 EU/mg

¹ RS-3 lot 6-20-1x was obtained by 5x dilution of the lot RS-3 6-20-5x (4.2 mg/mL) with sterile PBS

² 1x without calcium and magnesium, pH 7.4 ± 0.1 (Corning catalog # 21-040-CM)

³ Determined by RP-HPLC using entolimod, a parental protein, as calibration standard

⁴ Visual assessment

⁵ Confirmed by DNA sequencing of the plasmid harboring GP532 gene

⁶ Calculated based on GP532 sequence

⁷ Predicted by ExPASy ProtParam based on GP532 sequence

CMC

Date

Study Director

Date

640 Ellicott St, Suite 444, Buffalo, NY 14203

Figure 1. SDS-PAGE analysis of GP532 RS-3 lot 6-20.

Lanes 1 and 2: GP532 5 µg and 10 µg load, respectively; lane 3: MW markers.

SHIMADZU LabSolutions Analysis Report

<Sample Information>

Sample Name : Control_RS-3_1X_dil.1-3
 Sample ID : 1
 Data Filename : ZORBAX-C8_FA_35-42-60_20min_Control_RS-3_1X_dil.1-3_Run 7.03.19_A.lcd
 Method Filename : GP532_ZORBAX300SB-C8_35-42-60_20min.lcm
 Batch Filename :
 Vial # : 1-1 Sample Type : Unknown
 Injection Volume : 20 µL
 Date Acquired : 7/3/2019 2:07:59 PM Acquired by : kurnasov
 Date Processed : 7/3/2019 3:25:50 PM Processed by : kurnasov

<Chromatogram>

<Peak Table>

Peak#	Ret. Time	Area	Height	Conc.	Unit	Mark	Name
1	8.811	949321	24811	100.000		M	
Total		949321	24811				

A.

B.

Figure 2. RP HPLC analysis of RS-3 lot 6-20.

Panel A: full profile; Panel B: expanded profile. Injection: 10 µg in 20 µL; Mobile phase A: 0.1% TFA in water; Mobile phase B: 0.1% TFA in acetonitrile; Flow rate: 0.8 mL/min; Gradient: 35 – 50% B in 50 min. HPLC system: Shimadzu; RP HPLC column: 4.6 x 250 mm 300SB-C8 (Zorbax).

640 Ellicott St, Suite 444, Buffalo, NY 14203

<Sample Information>

Sample Name : Standard_532_RS-3_Lot 6.20_C-0.5
Sample ID : 1
Data Filename : TSKgelNew_Standard_532_RS-3_Lot 6.20_C-0.5_Run 8.12.19.lcd
Method Filename : TSKgel_G2000SW-xl_GF.lcm
Batch Filename :
Vial # : 1-1 Sample Type : Unknown
Injection Volume : 20 uL
Date Acquired : 8/12/2019 6:02:58 PM Acquired by : kurnasov
Date Processed : 8/13/2019 7:17:10 PM Processed by : kurnasov

<Chromatogram>**<Peak Table>**

Peak#	Ret. Time	Area	Height	Conc.	Unit	Mark	Name
1	8.146	5262471	173165	100.000		M	
Total		5262471	173165				

Figure 3. SE HPLC analysis of RS-3 lot 6-20.

Injection: 10 µg in 20 µL; Mobile phase: 0.1 M Na-phosphate buffer pH 6.8, 0.1 M Na₂SO₄; Flow rate: 1 mL/min; HPLC system: Shimadzu; SE HPLC column: TSK GEL G2000 SWXL 30 x 7.8 mm (Sigma, #808540)

REVIEWERS' COMMENTS:

Reviewer #1 (Remarks to the Author):

I am happy with the authors responses. In an ideal world it would be nice to have some additional information on the questions I raised earlier. But I understand the author's constraints and I feel that their analysis and revisions have addressed my questions in a satisfactory mater. I very much appreciate the tremendous amount of work that went into this study and strongly support acceptance.

Reviewer #2 (Remarks to the Author):

The authors addressed the comments of both reviewers and provided additional data (stability) or a convincing explanation as to why they could not provide additional data (antibody classes). The revised manuscript can be accepted for publication.